

# A machine learning approach for identifying anatomical biomarkers of early mild cognitive impairment

Alwani Liyana Ahmad[1,2,3], Jose M. Sanchez-Bornot[4], Roberto C. Sotero[5], Damien Coyle[6], Zamzuri Idris[2,3,7] and Ibrahima Faye[1,8]

[1] Department of Fundamental and Applied Sciences, Faculty of Science and Information Technology, Universiti Teknologi PETRONAS, Seri Iskandar, Perak, Malaysia

[2] Department of Neurosciences, Hospital Pakar Universiti Sains Malaysia, Kubang Kerian, Kelantan, Malaysia

[3] Brain and Behaviour Cluster, School of Medical Sciences, Universiti Sains Malaysia, Kubang Kerian, Kelantan, Malaysia

[4] Intelligent Systems Research Centre, School of Computing, Engineering and Intelligent Systems, Ulster University, Magee Campus, Derry Londonderry, United Kingdom

[5] Department of Radiology and Hotchkiss Brain Institute, University of Calgary, Calgary, Alberta, Canada

[6] The Bath Institute for the Augmented Human, University of Bath, Bath, United Kingdom

[7] Department of Neurosciences, School of Medical Sciences, Universiti Sains Malaysia, Kubang Kerian, Kelantan, Malaysia

[8] Centre for Intelligent Signal & Imaging Research (CISIR), Universiti Teknologi PETRONAS, Seri Iskandar, Perak, Malaysia

Corresponding author
Ibrahima Faye,
ibrahima_faye@utp.edu.my

## ABSTRACT

**Background.** Alzheimer's Disease (AD) poses a major challenge as a neurodegenerative disorder, and early detection is critical for effective intervention. Magnetic resonance imaging (MRI) is a critical tool in AD research due to its availability and cost-effectiveness in clinical settings.

**Objective.** This study aims to conduct a comprehensive analysis of machine learning (ML) methods for MRI-based biomarker selection and classification to investigate early cognitive decline in AD. The focus to discriminate between classifying healthy control (HC) participants who remained stable and those who developed mild cognitive impairment (MCI) within five years (unstable HC or uHC).

**Methods.** 3-Tesla (3T) MRI data from the Alzheimer's Disease Neuroimaging Initiative (ADNI) and Open Access Series of Imaging Studies 3 (OASIS-3) were used, focusing on HC and uHC groups. Freesurfer's recon-all and other tools were used to extract anatomical biomarkers from subcortical and cortical brain regions. ML techniques were applied for feature selection and classification, using the MATLAB Classification Learner (MCL) app for initial analysis, followed by advanced methods such as nested cross-validation and Bayesian optimization, which were evaluated within a Monte Carlo replication analysis as implemented in our customized pipeline. Additionally, polynomial regression-based data harmonization techniques were used to enhance ML and statistical analysis. In our study, ML classifiers were evaluated using performance metrics such as Accuracy (Acc), area under the receiver operating characteristic curve (AROC), F1-score, and a normalized Matthew's correlation coefficient (MCC′).

**Results.** Feature selection consistently identified biomarkers across ADNI and OASIS-3, with the entorhinal, hippocampus, lateral ventricle, and lateral orbitofrontal regions being the most affected. Classification results varied between balanced and imbalanced datasets and between ADNI and OASIS-3. For ADNI balanced datasets,
the naíve Bayes model using $z$-score harmonization and ReliefF feature selection performed best (Acc = 69.17%, AROC = 77.73%, F1 = 69.21%, MCC' = 69.28%). For OASIS-3 balanced datasets, SVM with zscore-corrected data outperformed others (Acc = 66.58%, AROC = 72.01%, MCC' = 66.78%), while logistic regression had the best F1-score (66.68%). In imbalanced data, RUSBoost showed the strongest overall performance on ADNI (F1 = 50.60%, AROC = 81.54%) and OASIS-3 (MCC' = 63.31%). Support vector machine (SVM) excelled on ADNI in terms of Acc (82.93%) and MCC' (70.21%), while naïve Bayes performed best on OASIS-3 by F1 (42.54%) and AROC (70.33%).

**Conclusion**. Data harmonization significantly improved the consistency and performance of feature selection and ML classification, with $z$-score harmonization yielding the best results. This study also highlights the importance of nested cross-validation (CV) to control overfitting and the potential of a semi-automatic pipeline for early AD detection using MRI, with future applications integrating other neuroimaging data to enhance prediction.

ta harmonization

# INTRODUCTION

Alzheimer's disease (AD) is marked by the gradual accumulation of amyloid-$\beta$ (amyloid plaques) in the extracellular space and tau proteins (neurofibrillary tangles–NFT) in the intracellular space of a neuron, leading to cognitive and motor dysfunctions and difficulties in daily activities. The symptomatic onset of AD is gradual, beginning with losses in episodic and semantic memory, progressing to aphasia, apraxia, mood disturbances, and more severe symptoms in the advanced stages (*Frisoni & Weiner, 2010*; *Petrella, Coleman & Doraiswamy, 2003*). Post-mortem examinations reveal patterns of neurodegeneration in brain regions corresponding to these cognitive and behavioral changes, as delineated by Braak's staging (*Braak et al., 2006*). The medial temporal lobe (MTL), including the hippocampus, amygdala, and entorhinal cortex, undergoes significant atrophy, which impacts memory formation and consolidation. Interestingly, early changes are also observed in the limbic system, encompassing the hippocampus, amygdala, cingulate, and parahippocampal gyri, affecting emotion and memory processing. The limbic system is connected to the entorhinal cortex *via* the subiculum, through which it is hypothesized that AD pathology spreads from one region to adjacent ones (*Didic et al., 2011*). However, *Braak & Del Tredici (2015)* reported that the very-early AD changes can be observed in the transentorhinal region in stage I when prospective AD patients remain asymptomatic, and from there, it spreads to the entorhinal region and the hippocampal formation in stages II and III, respectively. Therefore, when patients have the first symptoms of AD, they may be already in an irreversible stage. As AD advances, further anatomical changes include

atrophy in association cortical areas and ventricular enlargement (*Thompson et al., 2003*; *Apostolova et al., 2007*; *Nestor et al., 2008*).

The cascade of anatomical changes can be observed *in vivo* using neuroimaging and clinical data, *e.g.,* using positron emission tomography (PET) and cerebrospinal fluid (CSF) analysis to detect abnormal accumulation of amyloid plaques and tau proteins in the brain (*Petrella, Coleman & Doraiswamy, 2003*; *Faull et al., 2014*; *Apostolova, 2016*). Additionally, single-photon emission computed tomography (SPECT), utilizing a ligand binding to the dopamine transporter molecule (DaTscan), aids in evaluating Parkinsonian syndrome and distinguishing it and Lewy Body dementia from AD (*De la Fuente-Fernández, 2012*; *Sullivan et al., 2012*; *Papathanasiou et al., 2012*; *Magesh, Myloth & Tom, 2020*). Researchers have also explored combining multiple neuroimaging modalities, including SPECT, PET, magnetic resonance imaging (MRI), functional MRI (fMRI), and magneto/electro-encephalography (M/EEG) (*Liu et al., 2015b*; *Liu et al., 2015a*), and integrating neuroimaging data with cognitive or clinical measurements (*Mofrad et al., 2021*; *Liu et al., 2022*). However, it is essential to recognize that while PET and SPECT provide valuable insights, they are more invasive, costlier, and less globally accessible than MRI scans (*Sullivan et al., 2012*; *Wernickand & Aarsvold, 2004*). Essentially, used alone or combined with other neuroimaging data, MRI remains indispensable for evaluating suspected dementia cases, and ruling out alternative causes such as microinfarcts and white matter lesions (*Sullivan et al., 2012*; *Chouliaras & O'Brien, 2023*; *Harper et al., 2013*). Also, the enhanced resolution of MRI images allows the quantification of regional cerebral atrophy, making it relevant for early dementia assessment despite its limitations (*Sullivan et al., 2012*; *Chouliaras & O'Brien, 2023*; *Beltrán et al., 2020*; *Salvatore et al., 2015*; *Harper et al., 2015*; *Harper et al., 2016*; *Yue et al., 2018*; *Risacher et al., 2009*).

On the other hand, it has been found that pathogenic infections like prions have a significant impact on the neuronal atrophy and disruption of connectivity hubs within the medial temporal lobe (*Rábano et al., 2021*), leading to the hypothesis that AD could be triggered by the presence of a non-endogenous pathogen (*Braak & Del Tredici, 2015*). This observation also relates to the AD's disconnection syndrome hypothesis (*Smailovic et al., 2020*; *Delbeuck, Collette & Vander Linden, 2007*). In particular, *Xiaoshu et al. (2016)* identified that damage to white and gray matter within these regions disrupts limbic system networks, correlating with memory and behavioral impairments in AD patients. This disruption has been evidenced in neuroimaging studies using diffusion tensor imaging (DTI), MRI, and fMRI data (*Xiaoshu et al., 2016*; *Talwar et al., 2021*; *Kehoe et al., 2015*). However, minor fluctuations in behavior and emotional states can also be due to changes in diet (*Mohatar-barba & Fern, 2020*), lifestyle (*Tang et al., 2012*) or other less controlled factors, therefore posing a challenge in diagnosing mild cognitive impairment (MCI), a prodromal AD stage, and its progression to AD (*Moradi et al., 2015*; *McCombe et al., 2022*). This has led to a growing focus on developing automated diagnostic tools, primarily leveraging ML methods with neuroimaging data, for cost-effective and less subjective cognitive assessment (*Magesh, Myloth & Tom, 2020*; *Liu et al., 2015b*; *Beltrán et al., 2020*; *Salvatore et al., 2015*; *Harper et al., 2015*; *Moradi et al., 2015*; *McCombe et al., 2022*; *Klöppel et al., 2012*).

ML is increasingly utilized in healthcare for early-stage disease diagnosis, including cancer (*Yue et al., 2018*; *Kourou et al., 2015*; *Cruz & Wishart, 2006*; *Amrane, 2018*) and AD (*Risacher et al., 2009*; *Lebedeva et al., 2017*; *Vaghari et al., 2022*; *Islam & Zhang, 2018*; *Li et al., 2020*), reducing the possible subjectivity of diagnostic outcomes. However, AD research often focuses on comparing AD *vs.* healthy control (HC) participants data or using data from MCI participants who are already in an irreversible or progressive stage, potentially overlooking the early AD stage (*Risacher et al., 2009*; *Echávarri et al., 2011*; *Sun et al., 2018*; *Albert et al., 2018*; *Alderson et al., 2017*; *Garg et al., 2022*). Interestingly, *Popuri et al. (2020)* trained a classifier to discriminate between HC and AD participants using MRI data and posteriorly applied this classifier to predict MCI conversion to AD in 6 months or more, with an area under the receiver operating characteristic curve (AROC) of 0.81 for six months conversion and 0.73 for seven years conversion. This study also demonstrated the advantages of using data harmonization, *e.g.*, removing the data variability due to nuisance variables such as age, sex, and intracranial volume (ICV), for increasing classifier performance. Although not considered in our study, *Ma et al. (2019)* also compared different data harmonization strategies, including three different methods for ICV calculation, and their impact on classification performance. As reported in this study, data harmonization can improve the results as variability in the post-processed data can be more exclusively associated with changes due to AD progression.

Moreover, combining different techniques with classification methods has also helped improve the prediction outcome, as demonstrated by applying graph analysis tools with support vector machine (SVM) (*Kecman, 2005*) for predicting the risk of dementia among MCI patients in an EEG study (*Rossini, Miraglia & Vecchio, 2022*). Nevertheless, it is critically important to properly evaluate different methodologies to ensure reproducibility and potential implementation for clinical applications. For example, based on a Monte Carlo simulation data analysis, *Stamate et al. (2019)* introduced an ML framework to compare multiple classification models and found that the top-performing methods for predicting dementia and MCI were based on decision trees algorithms and the eXtreme Gradient Boosting model with the ReliefF (*Urbanowicz et al., 2018*; *Robnik-Šikonja & Kononenko, 2003*) method applied for feature selection. Significantly, the evaluation and comparison among different classification methods often rely on the performance of the classification accuracy, although this statistic may be biased for analysis involving imbalanced data (*Douzas, Bacao & Last, 2018*; *Chawla, Japkowicz & Kotcz, 2004*). In the medical field, imbalanced datasets are very common because of the lower number of abnormal cases compared to normal cases. This situation leads to misclassification for cases in the minority group, which may hamper the research on early AD detection (*Rahman & Davis, 2013*).

Addressing imbalanced data, various methods have been proposed which mainly combine resampling techniques with cost-sensitive classification approaches (*Ling & Sheng, 2008*). For example, *Chawla et al. (2002)* introduced an oversampling technique known as Synthetic Minority Over-sampling Technique (SMOTE), which was demonstrated in combination with a C4.5 decision tree and Ripper (*Chawla et al., 2002*; *Cohen, 1995*)

and naïve Bayes classifiers. In contrast, *Rahman & Davis (2013)* explored different under-sampling strategies as alternatives to SMOTE. So far, in the literature on imbalance data classification, RUSBoost is one of the most successful classification methods, combining under-sampling and boosting algorithms (*Seiffert et al., 2010*; *VanHulse, Khoshgoftaar & Napolitano, 2007*). However, in general, both under-sampling and over-sampling techniques present advantages and limitations, *e.g.*, whereas over-sampling methods increase the computational time and risk of overfitting due to sample duplication, mainly for the minority class, under-sampling may incur data loss, mainly for the majority class (*Drummond & Holte, 2003*).

Our study investigates early MRI-based anatomical changes linked to cognitive decline. Essentially, we propose a ML framework combining nested cross-validation (CV) with Bayesian optimization, as evaluated within a Monte Carlo replication analysis, to ensure the stability and reproducibility of the findings and a comprehensive evaluation, as demonstrated with balanced and imbalanced datasets. We used the normalized Matthew's correlation coefficient (MCC') and F1-score (*Boughorbel, Jarray & El-Anbari, 2017*; *Chicco & Jurman, 2020*), besides accuracy and AROC statistics, to more fittingly evaluate ML classifiers performance. For analyzing the early AD anatomical changes, we assessed the brain regional atrophy using ADNI and OASIS-3 datasets while examining a subset of HC participants who remained stable during these respective studies, in contrast to those participants who converted to MCI in less than 5 years. The analyzed MRI images for both groups were recorded at baseline, where all the participants were healthy. The Freesurfer software (*Fischl, 2012*) was used for the semi-automatic processing of the MRI data. Our approach also evaluated the possible advantages of data harmonization while comparing various feature selection and ML classification methods on different dataset cohorts: first, using MATLAB's Classification Learner (MCL) app, and then using our proposed ML framework. The main findings showed anatomical changes in MTL brain regions associated with potential cognitive decline, which align well with previous reports and were consistently found across the application of multiple feature selection and ML methods.

## MATERIALS AND METHODS

Portions of this text were previously published as part of a preprint (https://doi.org/10.48550/arXiv.2407.00040). Our methodology involves five key steps: (1) Data selection of participants from ADNI and OASIS-3 datasets who remained healthy during the study (HC) and those progressing to MCI over five years (uHC), producing imbalanced datasets (Fig. 1A). (2) Data processing was optionally used for each data to reduce variability due to sex, age, and ICV, using the HC group as a reference. Two different approaches are evaluated: residual and $z$-score harmonization (Fig. 1B). (3) The SPSS statistical software (*Morgan et al., 2019*; *Morgan et al., 2004*) was used to perform feature selection analyses for uncorrected and harmonized features using the ADNI imbalanced dataset (Fig. 1C). (4) The MCL app was used to evaluate different classification and feature selection methods based exclusively on a randomly selected ADNI-balanced cohort to select the

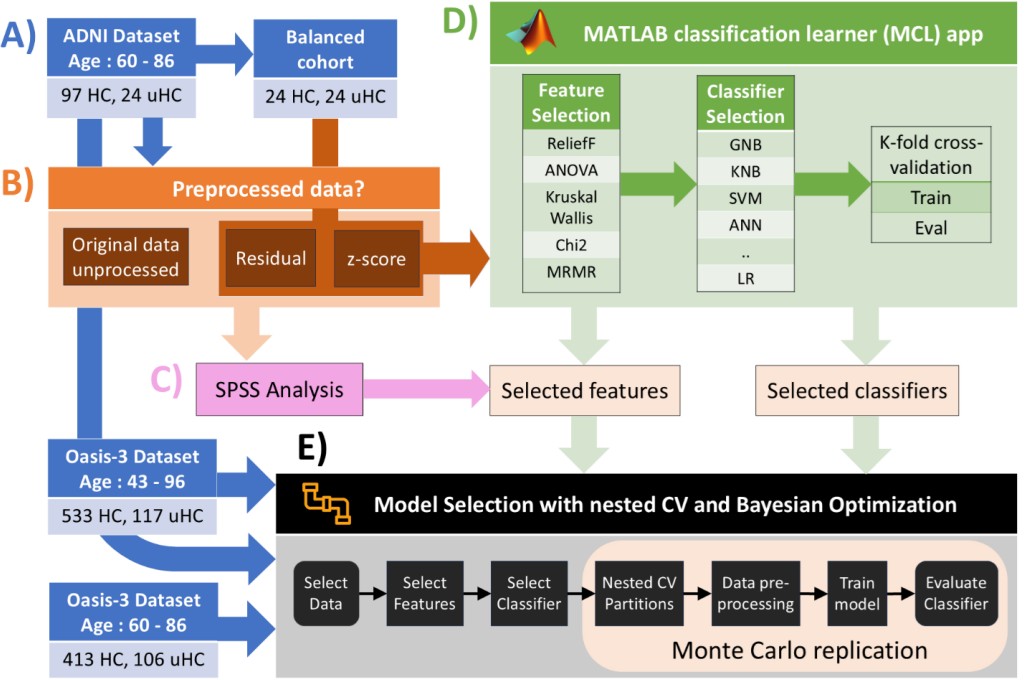

**Figure 1 Workflow illustrating the proposed methodology.** (A) Selection of participants data corresponding to healthy controls (HC) and participants who transitioned to MCI (uHC) in a period lower or equal than 5 years for ADNI and OASIS-3 datasets. These are imbalanced datasets as shown by the integer values indicating the number of samples in each group. A manually balanced cohort was extracted from ADNI dataset to be used within MCL app analysis. (B) Each data was optionally pre-processed using two different data correction procedures: residual and z-score harmonization. (C) Uncorrected and processed ADNI data undergone statistical analysis using SPSS software for assessing significant features. (D) The MATLAB's Classification Learner (MCL) app was utilized for evaluating a wide range of feature selection and classification methods, using an ADNI-balanced cohort. The MCL app includes many popular classifiers, such as Gaussian/Kernel Naïve Bayes (GNB/KNB), support vector machine (SVM), and artificial neural networks (ANN). Overall, we performed a preliminary selection of "best" classifiers and features from the MCL app and SPSS analysis. (E) Further evaluation of selected features and classification methods was performed through our proposed customized pipeline, implementing nested cross-validation (CV) and Bayesian optimization within a Monte Carlo replication framework. This last analysis was performed for both ADNI and OASIS-3 imbalanced datasets.* MATLAB symbol derived from: https://www.mathworks.com/?s_tid=gn_logo.

most appropriate approaches for posterior analyses (Fig. 1D). (5) Further evaluation of selected features and classifiers was performed through a customized pipeline, combining nested CV and Bayesian optimization within a Monte Carlo replication analysis (Fig. 1E). This pipeline enabled the implementation of imbalanced and balanced data analysis for ADNI and OASIS-3 datasets. For the latter analysis, balanced cohorts were generated from imbalanced datasets by randomly selecting the same number of samples in the majority as in the minority group. Lastly, performance metrics obtained from the pipeline calculations, such as F1 and Matthew's correlation coefficient (MCC) scores, were submitted to N-way ANOVA and multiple comparison analyses to evaluate the selected pipeline options.

## Participants data

We selected MRI data from two longitudinal studies: ADNI (http://adni.loni.usc.edu/) (*Jack Jr et al., 2008*) and OASIS-3 (*LaMontagne et al., 2019*) (http://www.oasis-brains.org). Both datasets are open source but require approval from their respective teams before access is provided. Please note that no data was collected during the implementation of this study. Participants data was collected in accordance with the Declaration of Helsinki for both studies, and procedures were approved by the local institutional Review Boards of participating centers in the ADNI study and following the guidelines of the Washington University Human Studies Committee in the OASIS-3 study, under approval number ADC-039.

The rationale behind using two different datasets is to compare and validate our methods with more heterogeneous data. Even when ADNI is already a multisite project, it follows a much stricter acquisition protocol than other studies. In summary, the ADNI study was launched in 2003 with the primary goal to test whether neuroimaging modalities such as MRI and PET can be analyzed independently or combined with other clinical and neuropsychological data to find Alzheimer's biomarkers and study the progression from HC to AD (https://adni.loni.usc.edu/methods/documents/). The OASIS-3 is a series of neuroimaging studies for which datasets are publicly available, as collected by the Knight Alzheimer Disease Research Center (ADRC) and its affiliated organizations (*Marcus et al., 2010*). Similarly to ADNI, OASIS-3 contains longitudinal data involving MRI and PET neuroimaging, as well as clinical, cognitive, and biomarker data from both normal aging and AD participants (*LaMontagne et al., 2019*; *Marcus et al., 2010*).

Subjects with an unavailable 3T MRI image at the baseline were excluded from this study. We specifically chose Magnetization Prepared RApid Gradient Echo (MP-RANGE) MRI images without repetition. We restricted our analysis to using only 3T MRI images from both datasets to simplify our study's complexity and ensure our results' consistency and reliability. 3T MRI scanners deliver a higher signal-to-noise ratio (SNR) and better spatial resolution than 1.5T scanners, resulting in higher image resolution (*Graves, 2022*). Furthermore, we avoided combining data from 1.5T and 3T MRI scanners as it could introduce variability due to differences in image acquisition protocols, and the differential analysis between the results for 3T and 1.5T analysis is beyond our present objectives. Additionally, it has been reported that changes in brain tissue texture detected by 3T MRI can lead to earlier AD diagnosis compared to 1.5T MRI (*Leandrou et al., 2020*). Moreover, we extracted essential demographic and cognitive information for our analysis from the ADNI and OASIS-3 datasets, including participants' sex, age, years of education, and Mini-Mental State Examination (MMSE) scores. Specifically, the ADNI participants selected for this study ranged in age from 60 to 86 and were either English or Spanish speakers.

The ADNI dataset is used in this work as the primary data for evaluation of ML classifiers during the analysis based on the MCL app. From the ADNI dataset, we selected 97 HC participants who remained stable during the study, as reflected in the ADNIMERGE table (downloaded from http://adni.loni.usc.edu/ in June 2022), which includes participants data for all the ADNI studies (ADNI-1, ADNI-GO, ADNI-2, and

ADNI-3). Additionally, we selected 24 participants who were diagnosed with HC at baseline and converted to MCI during a 5-year follow-up period after enrolling in the study. Otherwise, from the OASIS-3 dataset, we exclusively focused on MRI images for 533 HC and 117 uHC. Subjects in the OASIS-3 dataset were categorized according to the Clinical Dementia Rating (CDR). Participants with CDR =0 when their MRI image was first acquired and who remained stable during the study were considered HC. In contrast, participants who initially had a CDR of 0 but later showed an increase to a CDR of 0.5 at a subsequent visit were labeled uHC. For both data selected from ADNI and OASIS-3 datasets, the conversion period for uHC participants is 5 years or less from their first visit. We divided the OASIS-3 dataset into two cohorts based on age ranges: (1) the original participants' age range of 43–96 years and (2) a restricted age range of 60–86 years. The purpose of restricting the age range to 60–86 years is to match the ADNI dataset for comparison purposes, as structural brain changes depend on age (*Bethlehem et al., 2022*). Critically, MRI data for HC and uHC groups were all selected at baseline, where the participants were regarded as healthy. Table 1 summarizes the demographic information for selected participants in our study.

## MRI preprocessing pipeline

The MRI images were downloaded in NIFTI format and processed using FreeSurfer software (package version 7.3.2), with the standard cross-sectional pipeline *recon-all*, https://surfer.nmr.mgh.harvard.edu/fswiki/recon-all. In summary, this pipeline performs operations such as automatic co-registration to the Talairach atlas, image intensity normalization, and removal of non-brain tissue (*e.g.*, skull stripping) by utilizing a hybrid watershed/surface deformation procedure (*Se'gonne et al., 2004*), segmentation of grey matter (GM), white matter (WM), cerebrospinal fluid (CSF) tissues, subcortical brain regions automatic segmentation, and cortical automatic parcellation (*Fischl et al., 2004*; *Fischl et al., 2002*). The outcomes of the *recon-all* pipeline were carefully inspected to correct and ameliorate cortical and segmentation defects. Subsequently, the Freesurfer's *asegstats2table* and *aparcstats2table* scripts were run over this output, respectively, to extract the subcortical volume information tables for predefined regions and the different statistics (*e.g.*, volume and cortical thickness) for the cortical brain regions, which were extracted according to the Desikan atlas (*Desikan et al., 2006*). The ICV value was also estimated as part of the processing pipeline. Presumably, ICV provides a metric that resists change along aging for adults older than 50 years old, thus serving as a critical measure to control for brain size differences, for example, between female and male populations (*Ma et al., 2019*). Together with demographic information such as age and sex, using ICV can help remove unnecessary variation in the data that is not due to the brain degeneration process occurring in AD. In this study, we used only the brain volume information for the brain subcortical and cortical regions as extracted by the above MRI preprocessing pipeline. Moreover, we calculated total brain volumes by combining the values for the left and right hemispheres. In summary, we analyzed 39 merged brain volumes which were used as predictors in the ML analysis. Additionally, we included the

Ahmad et al. (2024), PeerJ, DOI 10.7717/peerj.18490

**Table 1 Demographic and clinical information for ADNI and OASIS-3 dataset's participants.** Information is provided as mean (SD), with *p*-values for intergroup (HC *vs* uHC) difference, and percentage for the minority group (%min). This latter information is provided to highlight the unbalance in the imbalanced data analysis.

| | ADNI: age 60–86 | | | | OASIS-3: age 43–96 | | | | OASIS-3: age 60–86 | | | |
|---|---|---|---|---|---|---|---|---|---|---|---|---|
| | HC | uHC | *p*-value | % min | HC | uHC | *P*-value | % min | HC | uHC | *p*-value | % min |
| Number of subjects | 97 | 24 | NA | 24.74 | 533 | 117 | NA | 21.95 | 413 | 106 | NA | 25.67 |
| Gender (M/F) | 56/41 | 12/12 | 0.686 | NA | 222/310 | 58/59 | 0.117 | NA | 175/238 | 53/53 | 0.159 | NA |
| Age (years) | 72.91 (5.96) | 75.95 (5.79) | 0.026 | NA | 66.71 (8.97) | 76.43 (7.40) | <0.001 | NA | 69.81 (5.11) | 76.08 (5.48) | <0.001 | NA |
| MMSE | 29.19 (1.12) | 28.67 (1.47) | 0.056 | NA | 29.20 (1.07) | 28.30 (1.61) | <0.001 | NA | 29.11 (1.11) | 28.31 (1.64) | <0.001 | NA |
| Years of education | 16.56 (2.39) | 16.00 (2.72) | 0.318 | NA | 16.38 (2.39) | 15.62 (2.90) | 0.003 | NA | 16.40 (2.37) | 15.65 (2.97) | 0.006 | NA |

measurement of brain segmentation volume without ventricles (BrainSegVolNotVent), bringing the total to 40 predictors.

## Data harmonization to eliminate the effects of nuisance factors

The purpose behind employing data correction is to eliminate the uncontrolled effect of nuisance factors on extracted brain regional measures, such as the effects of age, sex, and ICV; therefore, harmonized data would be less dependent on these variables, and thus we can assume that the main source of variability and differences among the HC and uHC harmonized data are due to the AD degenerative process. For example, it has been observed that brain structures vary across the lifespan, even in healthy aging, with non-linear and non-monotonic trajectories, although the trajectories become more linear for adults older than 50 years (*Bethlehem et al., 2022*). Typically, males have a larger average ICV than females, and brain regional volumes are correlated to ICV. Consequently, it may be appreciated that after controlling by ICV, sex-based differences are less noticeable (*Ma et al., 2019*).

In general, applying a correction to remove the effect of these variables can increase the performance of statistical and ML analysis (*Popuri et al., 2020*). Here, complementarily to previous studies (*Popuri et al., 2020; Ma et al., 2019; Ledig et al., 2018; Koikkalainen et al., 2012*), we adopted a multivariate polynomial regression approach for data harmonization, using age, sex, and ICV as covariates and setting the HC group as reference (*i.e.*, using exclusively the HC data to fit the polynomial regression parameters). To illustrate the possible advantages of this procedure, we used the whole dataset from HC, MCI, and AD groups available in the ADNIMERGE table and the hippocampus volume as a region of interest, which is one of the central brain regions suffering atrophy due to AD effects.

Two different harmonization approaches are discussed here. The first approach uses the residuals after fitting the polynomial to the HC data, while the second approach relies on the $z$-score transform, implemented using the following formulations:

$$\widetilde{poly}_G = \underset{poly_G}{\text{argmin}} \left\{ \sum_{i=i}^{N} \left( y_i^{(HC,G)} - \text{fit}\left( poly, Age_i^{(HC,G)}, ICV_i^{(HC,G)} \right) \right)^2 \right\}$$

$$\hat{\mu}_i, \hat{\sigma}_i = \text{predint}\left( \widetilde{poly}_{G_i}, Age_i, ICV_i, \right)$$

$$x_i^{(1)} = y_i - \hat{\mu}_i \; and \; x_i^{(2)} = \left( y_i - \hat{\mu}_i \right) / \hat{\sigma}_i$$

Here, the polynomials were fitted separately for each sex, $G = \{Male, Female\}$, using the MATLAB "fit" function, where $\widetilde{poly}_G$ represents the best-fitted polynomial model. $y_i^{(HC,G)}$, $Age_i^{(HC,G)}$, and $ICV_i^{(HC,G)}$, represent the $i - th$ measures for each participant in the HC group, considered separately for each value of $G$, for the corresponding variables. $\hat{\mu}_i$ and $\hat{\sigma}_i$ are the polynomial interpolation's mean and standard deviation estimates, calculated with the MATLAB "predint" function, for each sample in the dataset. These are required to derive the harmonized samples $x_i^{(1)}$ and $x_i^{(2)}$, obtained for each corresponding procedure, called residual and $z$-score harmonization, respectively.

## Statistical analysis for feature selection

We used the IBM SPSS Statistics software, version 28.0.1.1(15), to perform a statistical analysis of all available structural volume features obtained from the feature extraction analysis with FreeSurfer (Fig. 1C). We conducted a study of covariance (ANCOVA) only for the uncorrected data for each brain feature while using age, sex, years of education, and ICV as covariates (*Sarica et al., 2018*). Additionally, we applied both ANOVA and the independent sample non-parametric test of Kruskal–Wallis for all uncorrected and harmonized data while controlling for participant sex, age, and ICV variables. We employed the Bonferroni correction to correct for multiple comparisons. For the ADNI dataset, features that exhibited significant differences with $p$-value $\leq 0.05$ across all three analyses (ANCOVA, ANOVA, and Kruskal–Wallis) were selected for further classification analysis. The same analysis was later applied to the OASIS-3 dataset, and consistency among the selected features was evaluated.

## Feature and classification model selection in the MCL app

We utilized the MCL app, a graphical user interface (GUI) that facilitates feature and model selection through the tuning of predefined classification models based on $K$-fold cross-validation, holdout, or resubstitution validation, for binary and multiclass problems (Fig. 1D). The utilization of this app in our study is intended to simplify the process of exploring, building, training, and evaluating classification models. Within the MCL app, we explored all the available algorithms, including decision trees, discriminant analysis, logistic regression (LR), naïve Bayes, support vector machines, nearest neighbors, kernel approximation, ensemble methods, and neural networks, combined with the available feature selection techniques. We evaluated all these methods using the default predefined architectures and hyperparameter values. For example, the MCL app includes predefined bilayered neural network (BNN) and wide neural network (WNN) architectures from the neural networks' family. The classifiers showing better performance were saved as MATLAB scripts, which were then tailored to be used within our customized ML pipeline for a more comprehensive analysis based on nested CV combined with Bayesian optimization.

Although the MCL app also includes generic classification models with hyperparameters, which can be tunable through Bayesian optimization, we preferred to run and evaluate only the predefined models, as tuning the generic models can be very intensive and make the preliminary exploration more complex. For example, the predefined BNN model consists of two hidden layers with 10 neurons in each layer, while WNN is predefined with a single hidden layer with 100 neurons. By default, the activation function is ReLU in both cases. In contrast, the MCL app includes an "optimizable" neural network model, which selection enables to tune the number of hidden layers (up to 3), the number of artificial neurons in each layer, and select from a subset of activation functions (*e.g.*, ReLU, tanh, sigmoid), among other available hyperparameters.

On the other hand, during the evaluation with the MCL app, we applied all the available combinations between feature selection and predefined classification methods. The feature selection procedure not only aids in reducing overfitting (*Johnson & Kuhn,*

*2013*), but also facilitates faster training and decreases model complexity, making interpretation easier. Due to scale differences, the scores are converted into percentages to make feature selection more straightforward. Mainly, the available feature selection methods in the MCL app are (https://uk.mathworks.com/help/stats/feature-selection-and-feature-transformation.html):

- *Minimum Redundancy Maximum Relevance (MRMR)*

The MRMR algorithm calculates the importance of predictor variables by maximizing the mutual information between predictor and response variables while minimizing the mutual information between predictors.

- *Chi-square (Chi2)*

Ranks the features based on the *p*-value derived from the chi-square test. The potential independence between each predictor variable and the response variable was assessed through a separate chi-square test for each variable. The scores are represented as -log(p)

- *ReliefF*

ReliefF is particularly effective for evaluating the significance of each feature in distance-based supervised models. It weighs the distance between observations in the same group and different groups calculated with respect to their projection on the feature subspace.

- *ANOVA*

Conducts individual one-way analysis of variance for each predictor variable, categorized by class, and subsequently prioritizes features ranking based on the *p*-value. The score is represented as -log(p).

- *Kruskal–Wallis*

Ranks the features based on the *p*-values derived from the Kruskal–Wallis's test. The scores are represented as -log(p).

Using the MCL app's GUI options, we select the options to split the data into train (80%) and test (20%) subsets and set $K = 10$ for cross-validation to train and evaluate each classifier after using one of the available feature selection criteria separately in successive runs. This process was repeated 10 times with different random partitions to average the results and ensure more stable outcomes. For each process, we recorded the classifiers that achieved the highest accuracy. This process aims to identify the best models and features for subsequent classification analysis. Ultimately, we exported the best-performing classifiers (those that appeared most frequently as top performers) to corresponding implementations in MATLAB functions. This allowed for further performance evaluation using balanced and imbalanced data analysis with our customized ML pipeline.

Moreover, combining the SPSS statistical analyses in the previous section with the feature selection analyses in the MCL app, we ultimately proposed the following four selection criteria (selected features under these criteria are referred to as subset A-D features later in our analyses, denoting each subset with the corresponding letter in the below list):

(A) Average score percentage from the MCL app analysis

We combined the four scores calculated with the MCL app (chi-square, ANOVA, Kruskal–Wallis, and ReliefF) to create an average score. The selected features are those with scores at or above the median value.

(B) ReliefF

We selected only the features with positive scores from the ReliefF feature selection method in the MCL app, as negative scores indicate features of lesser importance (*Robnik-Šikonja & Kononenko, 2003*).

(C) Frequent feature appearances from all feature ranking analysis

We selected the features that consistently appeared across all the explored feature selection approaches among those selected from the MCL app and SPSS analyses.

(D) Feature selection according to SPSS analysis

We selected the features with significant differences in the HC *vs.* uHC statistical analysis performed in SPSS (*e.g.*, combining ANOVA, ANCOVA, and Kruskal–Wallis outcome). Note that we exclusively used an ADNI-balanced cohort for this preliminary analysis (arrow path from Figs. 1A to 1D), randomly generated from the ADNI imbalanced dataset, since available MATLAB classifiers are primarily optimized for balanced data analysis. ADNI dataset adheres to a much stricter acquisition protocol and has been extensively used in numerous previous studies (*Bethlehem et al., 2022*; *Ledig et al., 2018*; *Feng et al., 2020*; *Grueso & Viejo-sobera, 2021*; *Pellegrini et al., 2018*), offering a more reliable basis for comparison than the OASIS-3 dataset.

Ultimately, our research emulates the case when the outcome of one study is attempted to be replicated in other studies using different datasets. Thus, we evaluated the preliminarily selected ML classifiers and features in a posterior analysis, through the application of our customized pipeline to analyze imbalanced and randomly balanced ADNI and OASIS-3 datasets. For the latter case, each selected classifier will be retrained for each subset of selected features for the different datasets, as part of the application of the pipeline.

## Classification performance metrics

To evaluate the performance in binary classification problems, we calculated several statistical scores for the different techniques in our study, such as accuracy (Acc), $F1$, and Matthew's correlation coefficient (MCC), also known as Yule's phi coefficient. The F1 and MCC scores are essentially recommended for imbalanced classification problems. However, the MCC score has been reported as superior in accuracy and F1 in binary classification problems (*Boughorbel, Jarray & El-Anbari, 2017*; *Chicco & Jurman, 2020*). For clarity and self-content reasons, we present these metrics as follows, based on the variables represented in Table 2:

**Table 2** **The contingency table illustrates the notation for the number of cases with an existing/absent condition (CE/CA) evaluated using a generic test procedure, resulting in positive/negative examination (EP/EN) cases.** Combining the Condition and Examination labels, the cases can be partitioned as true/-false positive (TP/FP) and true/false negative (TN/FN).

| | Examination | | |
| --- | --- | --- | --- |
| Condition | Positive | Negative | Total = CE + CA |
| Existing | TP | FN | CE |
| Absent | FP | TN | CA |
| Total = EP + EN | EP | EN | |

$$Acc = \frac{TP + TN}{Total},$$

$$F1 = \frac{2}{\frac{1}{PPV} + \frac{1}{TPR}} = \frac{2 * PPV * TPR}{PPV + TPR},$$

$$MCC = \sqrt{TPR * TNR * PPV * NPV} - \sqrt{(1 - TPR) * (1 - TNR) * (1 - PPV) * (1 - NPV)},$$

$$= (TP * TN - FP * FN) / \sqrt{CE * CA * EP * EN}$$

$$MCC' = 0.5 * (1 + MCC),$$

where $TPR$ and $TNR$ represent the true positive and negative rate, also known as sensitivity (recall) and specificity, respectively ($TPR = TP/CE$ and $TNR = TN/CA$). $PPV$ and $NPV$ represent the positive and negative predictive values, respectively ($PPV = TP/EP$ and $NPV = TN/EN$). The $PPV$ is also commonly known as precision.

The Acc and F1-scores are defined in the range $[0, 1]$, where a value near to 1 indicates an excellent performance. Otherwise, MCC is defined in the range $[-1, 1]$, reaching 1 for perfect classification, when $TP = CE = EP$, and $TN = CA = EN$, and reaching $-1$ for a completely wrong classification when $FN = CE = EN$ and $FP = CA = EP$. However, we prefer to use the normalized MCC (MCC′) score as it is equivalent to the original but defined in the range $[0, 1]$, which eases the visual comparison with the Acc and F1-scores.

## Further validation with a customized ML pipeline

After selecting the feature and classification approaches using the MCL app and SPSS tools for each data harmonization approach, we evaluated each method combination further with nested CV and Bayesian optimization within a Monte Carlo replication analysis (Fig. 1E). To implement nested CV, in the external loop, for each $k = 1, \ldots, K$ ($K = 10$), 10% of samples are left out as the holdout subset. Then, the optimal hyper-parameters are selected for each corresponding model using a MATLAB-based Bayesian optimization procedure, automatically implementing an internal $K - 1$ fold cross-validation. Here, the partitions were created using MATLAB's "cvpartition" function, taking into consideration the sample group information (HC or uHC). This guarantees that each partition has similar proportions in each group (stratified partitions), which is critical to ensure robustness in imbalanced data analysis. We replicated this procedure 20 times with a Monte Carlo analysis to obtain repeated measurements for the above metrics, enabling a statistical comparison analysis to assess the better combination of pipeline options.

Moreover, for the Bayesian optimization approach, we used 200 iterations to enable the algorithm to find the "optimal" configuration of hyperparameters for each corresponding classifier. Several optimizable options were selected among the available ones as follows:

(1) *Naïve Bayes* (https://uk.mathworks.com/help/stats/fitcnb.html)

Data distribution assumption: "normal" or "kernel".

Kernel smoother type: "box", "epanechnikov", "normal", or "triangle".

Kernel smoothing window width: unbounded positive real number.

(2) *K-nearest neighbors* (https://uk.mathworks.com/help/stats/fitcknn.html)

Number of neighbors: integer number, restricted for values in the range [5, 30].

Distance function: "cityblock", "chebychev", "correlation", "cosine", "euclidean", "hamming", "jaccard", "mahalanobis", "minkowski", "seuclidean", or "spearman".

(3) *SVM* (https://uk.mathworks.com/help/stats/fitcsvm.html)

Kernel function: "gaussian", "rbf", "linear", or "polynomial".

Kernel scale parameter: positive real value constrained in the range $[10^{-1}, 10]$.

Box constraint: positive real value constrained in the range $[10^{-1}, 10]$.

(4) *Logistic regression* (https://uk.mathworks.com/help/stats/fitclinear.html)

Lambda (logistic regression implemented with Lasso regularization): positive real value evaluated in the range $[10^{-3}, 10]$.

Score transformation: "none", "logit", "invlogit", or "doublelogit".

(5) *RUSBoost* (https://uk.mathworks.com/help/stats/fitcensemble.html)

Ensemble aggregation method: "RUSBoost".

Number of ensemble learning cycles: positive integer (unbounded).

Learning rate for shrinkage: positive real number defined in the range (0, 1].

Maximal number of decision splits: positive integer number (unbounded).

Firstly, our pipeline was directly applied to the imbalanced ADNI and OASIS-3 datasets to evaluate the different option combinations. Then, a similar analysis was performed for balanced data, which were randomly generated from the original ADNI and OASIS-3 imbalanced datasets within each Monte Carlo replication step, *i.e.,* by randomly undersampling the larger group to match the same number of samples as in the smaller group, before the evaluation of each method combination.

Finally, we performed a statistical analysis involving N-way ANOVA and pairwise comparisons, to assess the influence of the different options in our analyses, including the selection of harmonization, feature selection and classification combination. For the control of spurious outcomes due to multiple comparisons, we applied both the Bonferroni correction and the Benjamini–Hochberg method, which controls the false discovery rate (FDR). We also used the post hoc Tukey Honestly Significant Difference (HSD) test, assuming a significance threshold of *p*-value $\leq 0.05$ to identify statistically significant differences. For Benjamini–Hochberg method correction, we applied a 5% FDR correction.

## RESULTS

### Data correction to eliminate the nuisance factors

Figure 2 illustrates the data harmonization procedure using polynomial regression of hippocampal volumes for data extracted from the ADNIMERGE table for HC, MCI, and AD participants (see 'Materials and Methods''). The effect of harmonization is illustrated for the various subgroups, obtained from the combination of the diagnostic (HC, MCI, or AD), participants' sex (M –male, F –female), and three artificial subdivisions of the participants according to their ICV size (group ID = 0 for smaller ICV, ID = 1 for medium ICV, and ID = 2 for larger ICV), as identified in the legend inset (Fig. 2C). After harmonization, linear models were fitted for the corrected volumes for each subgroup as a function of participants' age to uncover the general data trends during the aging process.

As expected, the negative trend in uncorrected hippocampal volume is observed even for aging in healthy conditions. Fig. 2A shows that hippocampal volume data points for participants with larger ICV are primarily localized on the top ("+" marker). In contrast, hippocampal measures for smaller ICV are mainly localized on the bottom (" ×" marker), which exposes the positive correlation between ICV and hippocampal volume. It is also clear that the graphs for the linear fit of female hippocampal volume are lower than for male data for each diagnostic subgroup, reflecting that females have lower hippocampal volume on average. Moreover, the linear fit slopes are more similar except for the AD participants, where the slope is less negative for females than males (darker/brighter intensity for each color corresponds to the male/female data).

Figure 2B shows the differences among the combined subgroups for the harmonized data derived with the residual-data correction approach, equivalent to using the residuals from fitting the polynomial models for each sex separately (Fig. 2D). Similarly, Fig. 2C illustrates the changes observed from the second proposed harmonization procedure with the $z$-score-data correction, which uses the estimated mean and standard deviation at each interpolation point to calculate the $z$-scores (see 'Materials and Methods''). Data harmonization was utilized to remove the effects of sex, ICV, and age over the harmonized data. For both corrections, we observed that the slopes of the a-posteriori fitted linear models are near zero for each combined subgroup. At the same time, the differences between the males and females are more minor within each diagnostic subgroup. Noticeably, we can more easily appreciate that female AD participants at older ages have relatively larger hippocampal volumes than males after data harmonization. For male participants, the differences are more stable between diagnostic subgroups, *i.e.*, the slopes nearly remain the same regardless of the group. Figure 2D illustrates the polynomial interpolation surfaces, separately per sex, which are primarily linear except in the borders, where interpolation errors may increase due to scarcer points (more female/male data points for smaller/larger ICV and fewer points for the age range extremes). However, it can also be appreciated that there are apparent nonlinear local changes in the surfaces and slightly more curvature for males than females for the hippocampal volumes (Figs. 2D– 2E).

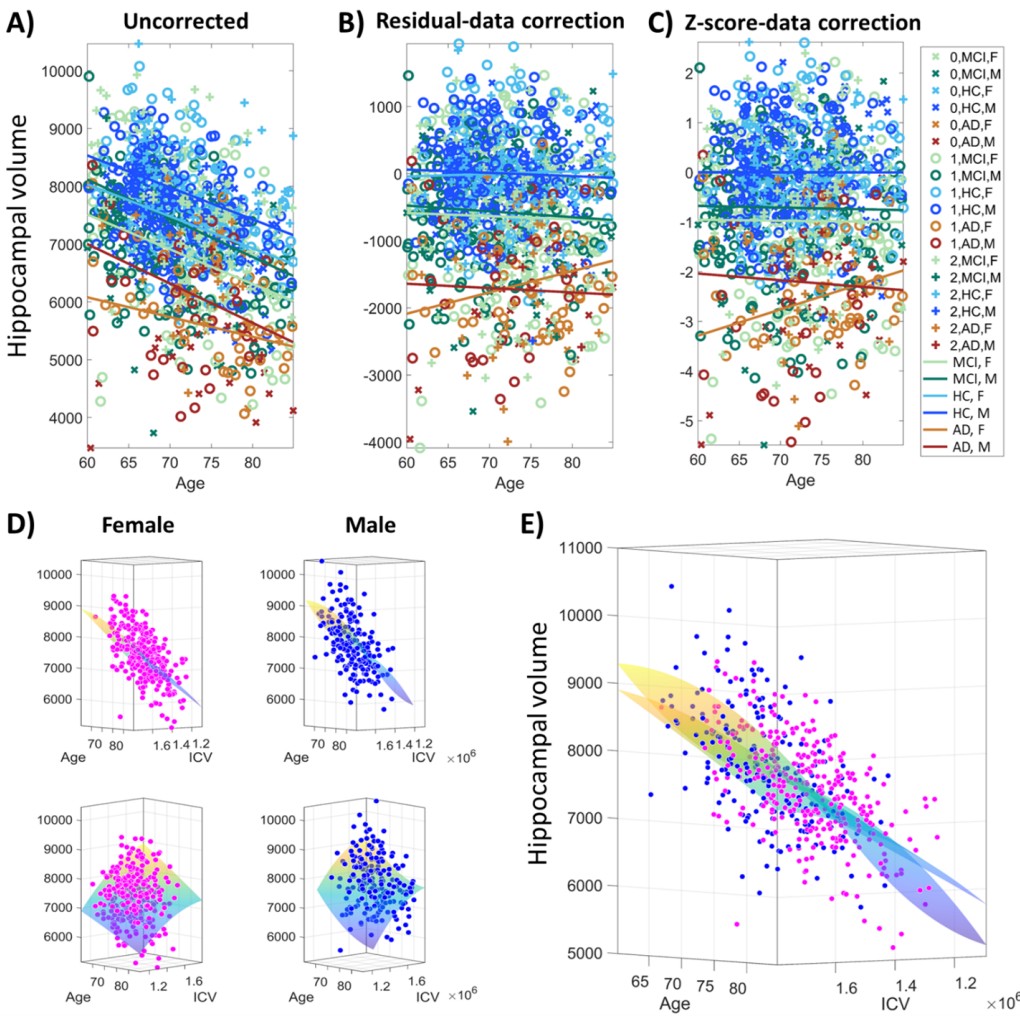

**Figure 2** **Data harmonization procedure illustrated for hippocampal volume variable in ADNIMERGE dataset.** Healthy Control (HC): females = 306, males = 213. Mild cognitive impairment (MCI): females = 219, males = 284. Alzheimer's disease (AD), females = 56, males = 76. (A) Original/uncorrected volume data as a function of age. (B) Data correction using linear regression fit's residuals (residual harmonization) with HC data as reference, calculated separately for female/male subgroups using age and intracranial volume (ICV) as covariates. (C) $z$-score correction using polynomial fit of degree (2,2) for interactions between age and ICV covariates, calculated separately for HC female/male subgroups. The mean and standard deviation of the polynomial fit in every point of the age-ICV subspace is used to calculate the $z$-score. (D) Illustration of the polynomial fitting procedure, separately for HC female/male subgroups. Two different views are illustrated in the top and bottom plots, same between the adjacent female and male plots. (E) Illustration of the polynomial fitting procedure for hippocampal volume for HC female and male subgroups data plotted together with the overlaid interpolating surface. The view is the same as in (D) top.

The calculated harmonized data (Figs. 2B–2C) can be used for statistical or classification analysis. For example, data harmonization may help to increase the statistical power necessary for variable selection to reduce dimensions before the classification analysis. Moreover, using higher-order polynomial regression may be advantageous
in better fitting the nonlinearity in the data. However, this may be an advantage only for larger datasets. In scenarios with a small amount of data, it is advisable to use linear interpolation, especially as the data fitting can be biased at the borders. We used polynomial fit (MATLAB script: 'poly22') only for illustrative purposes (based on hippocampal volume data). In the following analyses, we used linear interpolation (MATLAB script: 'poly11') to calculate both residual and $z$-score harmonization, considering all the features extracted from the Freesurfer's pipeline.

## Statistical analysis

Initially, we investigated early anatomical changes of AD based on the volumes of the Freesurfer-extracted brain regions for the uncorrected data, using ANOVA, Kruskal–Wallis, and ANCOVA tests in the SPSS statistical software. Whereas the ANCOVA analysis was performed for the uncorrected data for each brain feature, using age, sex, years of education, and ICV as nuisance variables, ANOVA and Kruskal–Wallis were directly applied to all the uncorrected and harmonized data. Table 3 shows that results are more significant for the harmonized data than the uncorrected data. From these analyses, eight features were consistently found to significantly differ between the HC and uHC groups for the ADNI imbalanced dataset. These eight features were selected for posterior analyses. In contrast, highlighted here only for comparison purposes, sixteen and twelve features were significantly different for the analyses involving the imbalanced OASIS-3 dataset for the original age range and ADNI age-matched participants, respectively. As expected, more significant results were obtained as the OASIS-3 datasets have a larger sample size (Table 1). Interestingly, the results demonstrate consistency across the datasets as the eight features found significant with the ADNI data analysis also showed significant results for the OASIS-3 cohorts. Overall, these analyses revealed some advantages of data harmonization, as the corresponding outcomes showed more substantial differences.

## Comparison between data harmonization approaches using the MCL app

Here, we performed a preliminary analysis to assess which harmonization procedure could offer superior performance for classification analysis using the MCL app for the ADNI-balanced cohort. Table 4 and Fig. 3 display the average performance of the different data harmonization procedures for top-performance classification methods. The best results were achieved for the residual harmonization procedure, with Kernel Naïve Bayes achieving an accuracy of 76.95% and AROC of 84.0% with comparable superior sensitivity and specificity results to other methods. Similarly, the results for the other classification methods were superior for this harmonization procedure except for Coarse Tree. Different classification methods, including SVM and LR, were also evaluated, but their results were inferior. This analysis produced better results for residual-corrected data. This may be because, for $z$-score harmonization, a smaller sample size may negatively impact the calculation of the $z$-scores, particularly at the borders of the data space.

Ahmad et al. (2024), *PeerJ*, DOI 10.7717/peerj.18490

Peer**J**

**Table 3  List of features that showed significant differences while controlling for multiple comparison using the Bonferroni's correction ($p \leq 0.05$), for ADNI and OASIS-3 imbalanced datasets. Volumes are reported as mean (SD) values for the original (uncorrected) measurements.** From left to right, volume information is presented for the significant features, followed by the results for ANOVA, Kruskal Wallis and ANCOVA tests, all for the uncorrected data, followed by the first two tests' results for the residual and $z$-score-harmonized data. For ANCOVA test for uncorrected data, the covariates were age, gender, years of education and ICV.

| Features | Uncorrected data | | | | | Residual | | Z-score | |
|---|---|---|---|---|---|---|---|---|---|
| | HC (original volumes, mm³) (SD) | uHC (original volumes, mm³) (SD) | ANOVA p-value | Kruskal Wallis p-value | ANCOVA p-value | ANOVA p-value | Kruskal Wallis p-value | ANOVA p-value | Kruskal Wallis p-value |
| | | | | ADNI: age 60–86 | | | | | |
| Lateral Ventricle | 32,077.74 (15,998.18) | 50,914.57 (26,158.63) | <0.001 | <0.001 | 0.001 | <0.001 | 0.003 | <0.001 | 0.002 |
| Inf-Lat-Vent | 1,165.48 (672.08) | 2,163.08 (1,471.34) | <0.001 | <0.001 | <0.001 | <0.001 | 0.002 | <0.001 | 0.002 |
| Hippocampus | 7,592.62 (837.74) | 7,213.36 (829.68) | *0.051 | *0.092 | 0.010 | 0.005 | 0.015 | 0.005 | 0.014 |
| Accumbens-area | 900.29 (165.10) | 761.66 (186.26) | <0.001 | 0.005 | 0.004 | 0.003 | 0.011 | 0.002 | 0.011 |
| Entorhinal | 3,582.20 (648.10) | 3,378.46 (719.29) | *0.191 | *0.246 | 0.010 | 0.007 | 0.010 | 0.007 | 0.011 |
| Lateral orbitofrontal | 13,717.41 (1,373.75) | 13,239.04 (1,379.935) | *0.140 | *0.123 | 0.002 | 0.002 | 0.003 | 0.002 | 0.002 |
| Middle temporal | 20,807.65 (2,386.15) | 20,222.25 (2,563.62) | *0.303 | *0.349 | 0.015 | 0.012 | 0.027 | 0.011 | 0.021 |
| BrainSegVolNotVent | 1,025,466.82 (103,015.64) | 1,011,076.79 (94,271.69) | *0.567 | *0.626 | 0.002 | <0.001 | 0.002 | <0.001 | 0.002 |
| | | | | OASIS-3: age 43–96 | | | | | |
| Lateral Ventricle | 27,870.93 (16,230.48) | 43,413.22 (23,753.60) | <0.001 | <0.001 | 0.006 | <0.001 | 0.032 | <0.001 | 0.032 |
| Inf-Lat-Vent | 1,103.52 (665.42) | 2,062.28 (1,464.24) | <0.001 | <0.001 | <0.001 | <0.001 | <0.001 | <0.001 | <0.001 |
| Hippocampus | 7,763.43 (868.24) | 6,996.65 (879.65) | <0.001 | <0.001 | <0.001 | <0.001 | <0.001 | <0.001 | <0.001 |
| Amygdala | 3,166.92 (464.77) | 2,821.13 (529.46) | <0.001 | <0.001 | <0.001 | <0.001 | <0.001 | <0.001 | <0.001 |
| Accumbens-area | 965.69 (192.28) | 806.29 (192.31) | <0.001 | <0.001 | <0.001 | <0.001 | <0.001 | <0.001 | <0.001 |
**Table 3** (*continued*)

| Features | Uncorrected data | | | | | Residual | | Z-score | |
|---|---|---|---|---|---|---|---|---|---|
| | HC (original volumes, mm³) (SD) | uHC (original volumes, mm³) (SD) | ANOVA *p*-value | Kruskal Wallis *p*-value | ANCOVA *p*-value | ANOVA *p*-value | Kruskal Wallis *p*-value | ANOVA *p*-value | Kruskal Wallis *p*-value |
| Entorhinal | 3,691.79 (689.42) | 3,401.94 (781.46) | <0.001 | <0.001 | <0.001 | <0.001 | <0.001 | <0.001 | <0.001 |
| Fusiform | 17,639.88 (2282.77) | 16,793.58 (2,541.72) | <0.001 | <0.001 | <0.001 | <0.001 | <0.001 | <0.001 | <0.001 |
| Inferior temporal | 19,586.25 (2,783.38) | 18,433.00 (2,929.76) | <0.001 | <0.001 | <0.001 | <0.001 | <0.001 | <0.001 | <0.001 |
| Isthmus cingulate | 4,657.37 (687.63) | 4,570.60 (685.20) | *0.217 | *0.180 | 0.049 | 0.014 | 0.020 | 0.013 | 0.018 |
| Lateral orbitofrontal | 13,572.41 (1,565.69) | 13,221.11 (1,591.28) | 0.029 | 0.026 | 0.040 | 0.005 | 0.008 | 0.006 | 0.009 |
| Medial orbitofrontal | 10,136.71 (1,151.98) | 9,957.09 (1,243.07) | *0.133 | *0.098 | 0.004 | <0.001 | 0.001 | <0.001 | 0.002 |
| Middle temporal | 20,397.11 (2,793.42) | 19,323.31 (2,741.79) | <0.001 | <0.001 | <0.001 | <0.001 | <0.001 | <0.001 | <0.001 |
| Para hippocampal | 2,830.92 (526.19) | 3,628.37 (556.32) | <0.001 | <0.001 | 0.004 | <0.001 | <0.001 | <0.001 | <0.001 |
| Superior temporal | 21,659.54 (2,508.39) | 20,609.39 (2,840.67) | <0.001 | <0.001 | 0.007 | <0.001 | <0.001 | <0.001 | <0.001 |
| Insula | 13,042.72 (1,545.59) | 12,915.63 (1,673.89) | *0.428 | *0.583 | 0.020 | 0.004 | 0.006 | 0.004 | 0.007 |
| BrainSegVolNotVent | 1,042,137.77 (109,896.64) | 1,006,024.23 (109,950.49) | 0.001 | 0.003 | <0.001 | <0.001 | <0.001 | <0.001 | <0.001 |
| OASIS-3: age 60–86 | | | | | | | | | |
| Inf-Lat-Vent | 1,181.78 (681.07) | 2,062.96 (1,463.92) | <0.001 | <0.001 | <0.001 | <0.001 | <0.001 | <0.001 | <0.001 |
| Hippocampus | 7,639.15 (801.46) | 7,013.75 (825.28) | <0.001 | <0.001 | <0.001 | <0.001 | <0.001 | <0.001 | <0.001 |
| Amygdala | 3,122.47 (430.58) | 2,825.05 (480.07) | <0.001 | <0.001 | <0.001 | <0.001 | <0.001 | <0.001 | <0.001 |
| Accumbens-area | 936.00 (173.92) | 805.50 (185.47) | <0.001 | <0.001 | 0.003 | <0.001 | <0.001 | <0.001 | <0.001 |
| Entorhinal | 3,673.07 (689.16) | 3,425.49 (785.00) | 0.001 | <0.001 | <0.001 | <0.001 | <0.001 | <0.001 | <0.001 |

Ahmad et al. (2024), *PeerJ*, DOI 10.7717/peerj.18490

Peer

**Table 3** (*continued*)

| Features | Uncorrected data | | | | | Residual | | Z-score | |
|---|---|---|---|---|---|---|---|---|---|
| | HC (original volumes, mm³) (SD) | uHC (original volumes, mm³) (SD) | ANOVA *p*-value | Kruskal Wallis *p*-value | ANCOVA *p*-value | ANOVA *p*-value | Kruskal Wallis *p*-value | ANOVA *p*-value | Kruskal Wallis *p*-value |
| Fusiform | 17,451.08 (2,175.02) | 16,908.48 (2,423.70) | 0.026 | 0.032 | 0.017 | <0.001 | <0.001 | <0.001 | <0.001 |
| Inferior temporal | 19,404.56 (2,752.38) | 18,550.90 (2,845.48) | 0.005 | 0.008 | <0.001 | <0.001 | <0.001 | <0.001 | <0.001 |
| Medial orbitofrontal | 10,102.50 (1,158.77) | 10,012.27 (1,198.65) | 0.478* | 0.453* | 0.019 | <0.001 | 0.004 | <0.001 | 0.004 |
| Middle temporal | 20,153.47 (2,723.91) | 19,491.38 (2,551.67) | 0.024 | 0.033 | 0.013 | <0.001 | <0.001 | <0.001 | <0.001 |
| Para hippocampal | 3,791.19 (516.83) | 3,639.69 (545.66) | 0.008 | 0.004 | 0.027 | 0.001 | 0.001 | 0.002 | 0.002 |
| Superior temporal | 21,337.81 (2,289.95) | 20,739.03 (2,575.37) | 0.020 | 0.014 | 0.051 | 0.002 | 0.003 | 0.003 | 0.004 |
| BrainSegVolNotVent | 1,031,148.72 (103,676.92) | 1,011,836.22 (104,411.90) | 0.088* | 0.153* | <0.001 | <0.001 | <0.001 | <0.001 | <0.001 |

**Notes.**

*An asterisk (*) denotes that the accompanying *p*-value is not significant.

From top to bottom, results are presented for ADNI and OASIS-3 datasets. Note that only the significant features obtained using the ADNI dataset were considered in posterior analyses. The results for the OASIS-3 datasets are only illustrated for comparison purposes.

**Table 4  Performance comparison between residual and z-score harmonization.**  The results are presented for some of the "top" classifiers as observed in the MCL app analysis for evaluated performance metrics: accuracy (Acc), sensitivity (Sen), specificity (Spec), and area under receiver operating characteristic curve (AROC). See Fig. 3 for complementary information.

| Model | Residual | | | | Z-score | | | |
|---|---|---|---|---|---|---|---|---|
| | Acc (%) | Sen (%) | Spec (%) | AROC (%) | Acc (%) | Sen (%) | Spec (%) | AROC (%) |
| Kernel Naïve Bayes | 76.95 | 76.84 | 77.11 | 84.0 | 71.80 | 67.43 | 76.05 | 76.0 |
| Cosine KNN | 75.23 | 70.17 | 80.00 | 78.0 | 64.10 | 47.37 | 80.00 | 65.0 |
| Coarse Tree | 74.40 | 75.00 | 73.68 | 74.0 | 76.90 | 80.00 | 73.68 | 77.0 |
| Ensemble: Bagged Trees | 76.90 | 73.68 | 80.00 | 77.0 | 66.65 | 63.16 | 70.00 | 67.0 |

## Features and classification methods evaluation

Using the MCL app, we also performed an exhaustive analysis to complement the previous analyses by evaluating and selecting the "best" feature selection and classification methods using the ADNI-balanced cohort. First, we calculated the percentages for the 40 features using all the feature selection methods available in the app. Then, we ignored the MRMR outcome, as only one feature (Inf-Lat-Vent volume) exhibited a score greater than 0. Table 5 reveals that the chi-square provides lower scores when compared to ANOVA and Kruskal–Wallis. Conversely, ANOVA and Kruskal–Wallis exhibited minimal discrepancies in their scores. This is the primary reason we converted the scores into percentages to enhance visual comparison and selection of the most relevant features. We also ranked the features based on how frequently each feature selection criteria selected them, combining the MCL app and SPSS analysis. As shown in Table 6, we found five features selected by all selection criteria. These features are BrainSegVolNotVent, Inf-Lat-Vent, entorhinal, lateral orbitofrontal, and lateral ventricle. Then, with a slightly lower rank, the parahippocampal and hippocampus regions were selected by 6/5 of the selection criteria.

Subsequently, we calculated the average for classification accuracy, sensitivity, specificity, and AROC statistics for each classifier and feature selection method for ten random replications. Table 7 reveals that Kernel Naïve Bayes was selected 34% of the time as the best-performance classifier, and its average accuracy was 77.3% by pooling together all the corresponding outcomes from the feature selection methods. Gaussian Naïve Bayes and Cosine KNN were tied up in second place, selected 7% of the time as the top performer, and with average accuracy performance of 73.05% and 71.5%, respectively. Among the other classifiers, the LR achieved an average accuracy of 75.65% with an AROC of 0.7592 when using the ReliefF method. However, it did not perform well for the other feature selection criteria. Regarding the best feature selection methods in the MCL app, ReliefF outperformed other methods (Fig. 4). It is important to emphasize that the outcomes from Tables 3–7 and Figs. 3–4 were derived from evaluation exclusively on the ADNI dataset. In the next section, we evaluate the generalization of these results using both ADNI and OASIS-3 datasets with our customized pipeline.

**Table 5  Score rating with percentage as calculated for the different feature selection methods as available in the MCL app.**

| Features | Chi-square | | ANOVA | | Kruskal–Wallis | | ReliefF | | Average score (%) Median = 1.79 |
|---|---|---|---|---|---|---|---|---|---|
| | Score | % Median = 1.94 | Score | % Median = 2.13 | Score | % Median = 1.94 | Score | % | |
| entorhinal | 2.28 | 3.80 | 5.48 | 6.07 | 5.48 | 6.35 | 0.04 | 13.50 | 7.43 |
| fusiform | 1.44 | 2.40 | 1.22 | 1.35 | 1.68 | 1.94 | 0.03 | 10.51 | 4.05 |
| Inf-Lat-Vent | 2.83 | 4.72 | 4.13 | 4.58 | 4.49 | 5.20 | 0.03 | 9.88 | 6.09 |
| temporalpole | 3.48 | 5.80 | 0.44 | 0.48 | 2.06 | 2.39 | 0.03 | 8.89 | 4.39 |
| posteriorcingulate | 1.44 | 2.40 | 0.68 | 0.75 | 0.92 | 1.07 | 0.02 | 7.11 | 2.83 |
| isthmuscingulate | 1.54 | 2.57 | 1.30 | 1.44 | 2.27 | 2.63 | 0.02 | 7.02 | 3.42 |
| Hippocampus | 1.14 | 1.90 | 4.29 | 4.75 | 4.14 | 4.80 | 0.02 | 6.70 | 4.54 |
| parahippocampal | 4.09 | 6.82 | 3.27 | 3.62 | 2.71 | 3.14 | 0.02 | 5.84 | 4.86 |
| parsopercularis | 1.34 | 2.23 | 1.79 | 1.99 | 1.75 | 2.03 | 0.02 | 5.62 | 2.97 |
| Lateral-Ventricle | 2.83 | 4.72 | 4.57 | 5.07 | 5.42 | 6.28 | 0.02 | 4.92 | 5.25 |
| transversetemporal | 0.63 | 1.05 | 0.02 | 0.02 | 0.26 | 0.30 | 0.01 | 4.70 | 1.52 |
| precentral | 0.70 | 1.17 | 3.81 | 4.22 | 3.81 | 4.42 | 0.01 | 4.45 | 3.56 |
| insula | 3.55 | 5.91 | 2.04 | 2.26 | 2.14 | 2.49 | 0.01 | 4.16 | 3.70 |
| frontalpole | 0.20 | 0.34 | 1.49 | 1.65 | 0.73 | 0.84 | 0.01 | 2.70 | 1.38 |
| superiortemporal | 1.39 | 2.32 | 0.28 | 0.31 | 0.36 | 0.42 | 0.00 | 1.52 | 1.14 |
| Amygdala | 2.77 | 4.61 | 0.87 | 0.95 | 1.16 | 1.35 | 0.00 | 1.17 | 2.02 |
| lateralorbitofrontal | 2.22 | 3.70 | 4.93 | 5.46 | 4.79 | 5.55 | 0.00 | 0.60 | 3.83 |
| lingual | 0.42 | 0.70 | 0.37 | 0.41 | 0.12 | 0.14 | 0.00 | 0.57 | 0.46 |
| BrainSegVolNotVent | 5.45 | 9.08 | 9.80 | 10.86 | 9.76 | 11.30 | 0.00 | 0.13 | 7.84 |
| parstriangularis | 0.48 | 0.81 | 0.22 | 0.25 | 0.30 | 0.35 | −0.00 | | 0.35 |
| middletemporal | 1.14 | 1.90 | 4.01 | 4.44 | 3.19 | 3.70 | −0.00 | | 2.51 |
| paracentral | 1.34 | 2.23 | 2.48 | 2.75 | 1.60 | 1.87 | −0.01 | | 1.71 |
| precuneus | 0.83 | 1.38 | 2.21 | 2.45 | 1.50 | 1.73 | −0.01 | | 1.39 |
| Accumbens-area | 1.44 | 2.40 | 4.12 | 4.56 | 3.65 | 4.23 | −0.01 | | 2.80 |
| medialorbitofrontal | 0.16 | 0.27 | 0.86 | 0.95 | 0.81 | 0.94 | −0.01 | | 0.54 |
| inferiorparietal | 0.42 | 0.70 | 1.61 | 1.78 | 1.20 | 1.39 | −0.01 | | 0.97 |
| superiorfrontal | 1.05 | 1.75 | 1.93 | 2.14 | 1.68 | 1.94 | −0.02 | | 1.46 |
| postcentral | 0.27 | 0.46 | 1.77 | 1.96 | 1.26 | 1.46 | −0.02 | | 0.97 |
| caudalmiddlefrontal | 0.48 | 0.81 | 1.97 | 2.18 | 1.98 | 2.30 | −0.02 | | 1.32 |
| lateraloccipital | 1.60 | 2.66 | 1.92 | 2.13 | 1.68 | 1.94 | −0.02 | | 1.68 |
| cuneus | 0.05 | 0.08 | 0.10 | 0.11 | 0.26 | 0.30 | −0.21 | | 0.12 |
| bankssts | 0.70 | 1.17 | 2.34 | 2.59 | 2.36 | 2.73 | −0.02 | | 1.62 |
| caudalanteriorcingulate | 1.19 | 1.98 | 1.17 | 1.30 | 1.01 | 1.17 | −0.02 | | 1.11 |
| parsorbitalis | 1.99 | 3.31 | 2.02 | 2.23 | 1.64 | 1.90 | −0.03 | | 1.86 |
| superiorparietal | 0.20 | 0.34 | 0.29 | 0.32 | 0.34 | 0.40 | −0.03 | | 0.26 |
| inferiortemporal | 1.00 | 1.67 | 2.49 | 2.76 | 1.64 | 1.90 | −0.03 | | 1.58 |
| rostralmiddlefrontal | 2.77 | 4.61 | 3.16 | 3.49 | 2.40 | 2.78 | −0.03 | | 2.72 |
| rostralanteriorcingulate | 1.05 | 1.75 | 3.43 | 3.80 | 2.86 | 3.30 | −0.03 | | 2.21 |
| pericalcarine | 1.14 | 1.90 | 0.35 | 0.39 | 0.22 | 0.25 | −0.03 | | 0.64 |
| supramarginal | 0.96 | 1.60 | 1.08 | 1.20 | 0.67 | 0.78 | −0.04 | | 0.89 |

**Table 6  Feature selection according to different selection criteria from MCL app and SPSS analysis.**

| Features | Chi-square | ReliefF | ANOVA | Kruskal–Wallis | Average score (MCL app): chi-square, ReliefF, ANOVA & Kruskal–Wallis | Statistical analysis (SPSS): ANOVA, ANCOVA & Kruskal–Wallis | Total |
|---|---|---|---|---|---|---|---|
| BrainSegVolNotVent | / | / | / | / | / | / | 6 |
| Inf-Lat-Vent | / | / | / | / | / | / | 6 |
| Entorhinal | / | / | / | / | / | / | 6 |
| Lateral orbitofrontal | / | / | / | / | / | / | 6 |
| Lateral Ventricle | / | / | / | / | / | / | 6 |
| Parahippocampal | / | / | / | / | / |   | 5 |
| Hippocampus |   | / | / | / | / | / | 5 |
| Accumbens-area |   |   | / | / | / | / | 4 |
| Middle temporal |   |   | / | / | / | / | 4 |
| Precentral |   | / | / | / | / |   | 4 |
| Insula | / | / |   |   | / |   | 3 |
| Temporal pole | / | / |   |   | / |   | 3 |
| Rostral middle frontal | / |   | / |   | / |   | 3 |
| Rostral anterior cingulate |   |   | / | / | / |   | 3 |
| Amygdala | / | / |   |   | / |   | 2 |
| Fusiform |   | / |   |   | / |   | 2 |
| Posterior cingulate |   | / |   |   | / |   | 2 |
| Isthmuscingulate |   | / |   |   | / |   | 2 |
| Parsopercularis |   | / |   |   | / |   | 2 |
| Parsorbitalis | / |   |   |   | / |   | 2 |
| Transverse temporal |   | / |   |   |   |   | 1 |
| Frontal pole |   | / |   |   |   |   | 1 |
| Superior temporal |   | / |   |   |   |   | 1 |
| Lingual |   | / |   |   |   |   | 1 |

## Balanced data analysis with customized pipeline

In the present and following sections, we further evaluate the selected "best" combination for classification methods, selected features, and data harmonization procedures with our customized pipeline for balanced and imbalance datasets, respectively. Five classifiers are compared in these analyses: naïve Bayes, KNN, SVM, LR, and RUSBoost. The purpose is to further evaluate the "best" candidates selected mainly from the above MCL app analysis, compared against RUSBoost, which is expected to show superior performance for imbalanced datasets. In contrast to the MCL app and feature selection analyses above, which could have had some bias due to the MCL app analysis being restricted to use a single balanced ADNI cohort, the current analyses are extended to include both the whole ADNI and OASIS-3 (original and ADNI-age matched) imbalanced datasets to evaluate the selected method combinations. Moreover, in this section, we performed randomly undersampling to balance these datasets within a Monte Carlo replication analysis, which subsequently runs our customized pipeline for each of the combined choices

**Table 7  Selection frequency as top performer for each classification method under different feature selection criteria.** The results for the classification methods are presented across the rows, whereas the columns present the outcome for the different feature selection strategies.

| Models | Chi-square | ReliefF | ANOVA | Kruskal–Wallis | Average score (MCL app): chi-square, ReliefF, ANOVA & Kruskal–Wallis | Statistical analysis (SPSS): ANOVA, ANCOVA & Kruskal–Wallis | Total appearances | Total (%) |
|---|---|---|---|---|---|---|---|---|
| Kernel Naïve Bayes | 7 | 7 | 6 | 6 | 6 | 2 | 34 | 34 |
| Gaussian Naïve Bayes | 1 | 2 | 1 | 1 | 2 | | 7 | 7 |
| Cosine KNN | 1 | 1 | 1 | | 1 | 3 | 7 | 7 |
| Logistic Regression Kernel | 1 | 2 | 2 | | | 1 | 6 | 6 |
| Weighted KNN | | | 1 | | | 3 | 4 | 4 |
| Subspace Discriminant | 3 | | 1 | | | | 4 | 4 |
| Subspace KNN | 1 | | 1 | 1 | | | 3 | 3 |
| Medium KNN | | | 1 | 1 | 1 | | 3 | 3 |
| SVM Kernel | 1 | 1 | | 1 | | | 3 | 3 |
| Linear SVM | 1 | 1 | | | | 1 | 3 | 3 |
| Fine tree | 1 | | | | 1 | 1 | 3 | 3 |
| Medium tree | 1 | | | | 1 | 1 | 3 | 3 |
| Coarse tree | 1 | | | | 1 | 1 | 3 | 3 |
| Linear Discriminant | 2 | | | | 1 | | 3 | 3 |
| Quadratic SVM | | 1 | 1 | | | | 2 | 2 |
| Trilayered Neural Network | | 1 | | | 1 | | 2 | 2 |
| Cubic KNN | | | | | | 2 | 2 | 2 |
| Fine KNN | | 1 | | | | 1 | 2 | 2 |
| Logistic Regression | | 1 | | | | | 1 | 1 |
| Ensemble: Subspace Discriminant | | | | | 1 | | 1 | 1 |
| Cubic SVM | | 1 | | | | | 1 | 1 |
| Medium Gaussian SVM | | 1 | | | | | 1 | 1 |
| Bagged Trees | 1 | | | | | | 1 | 1 |
| Ensemble: Bagged Trees | | | | | | 1 | 1 | 1 |

among four different groups of selected features (subsets A-D as shown in Table S13), five classifiers, three datasets, and two harmonization procedures. This last analysis may favor classifiers that perform better in balanced data scenarios, which can be compared against the following section results, where the same analysis will be applied without undersampling, *i.e.,* for the original imbalanced datasets. Evaluations are based on these performance metrics: Acc, AROC, F1, and MCC'. Interestingly, for balanced data analysis, metrics such as Acc and AROC serve as standards to evaluate the "best" classification performances, but this may not be the case for imbalanced data analysis, where F1 and MCC' are recommended (*Boughorbel, Jarray & El-Anbari, 2017*; *Chicco & Jurman, 2020*).

Figure 5 shows the results of the balanced data analysis. The best outcome for ADNI balanced cohorts was achieved using a Naïve Bayes classifier based on the ReliefF feature selection (subset B) and $z$-score data harmonization, achieving Acc $= 69.17 \pm 6.54\%$,

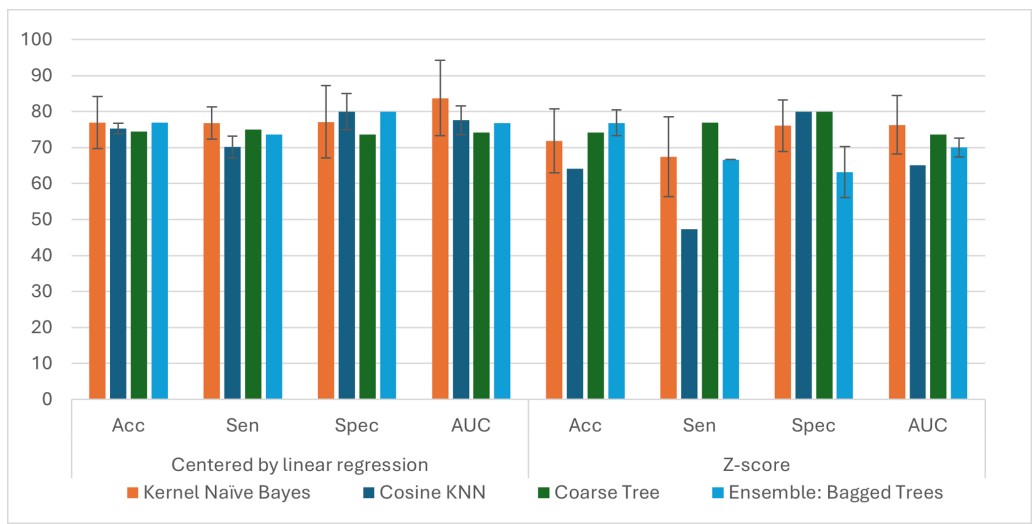

**Figure 3** **Comparison in classification between data harmonization procedures using linear regression based centered data (residual harmonization) and *z*-score harmonization.** The columns present the results for the top evaluated models, left-to-right order, according to the MCL app analysis, and column groups correspond to the different assessed performance metrics: accuracy (Acc), sensitivity (Sen), specificity (Spec), and area under curve of the ROC graph (AUC).

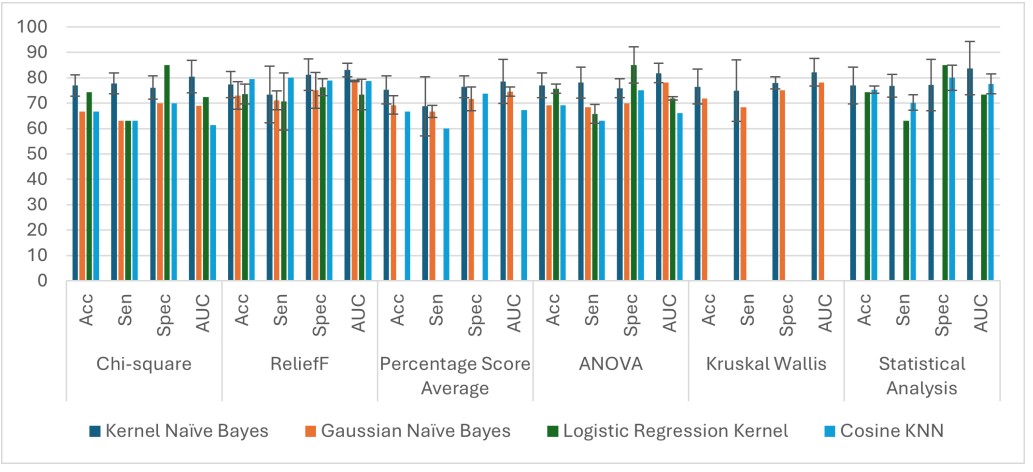

**Figure 4** **Average classification in model selection analysis.** Some bar plots do not display the standard error since the model was selected only once throughout the analysis.

AROC $= 77.73 \pm 7.08\%$, F1 $= 69.21 \pm 7.90\%$, and MCC' $= 69.28 \pm 6.56\%$ (FDR-adjusted *p*-value $p_{FDR} < 0.05$ for all multiple comparisons of Naïve Bayes *vs* other classifiers for each metric). However, this result was not replicated for the OASIS-3 cohorts, possibly revealing a selection bias as an individual balanced ADNI cohort was utilized in the previous MCL analysis for feature selection. For the OASIS-3 age-matched dataset, the best performance was obtained for the SVM classifier using the features

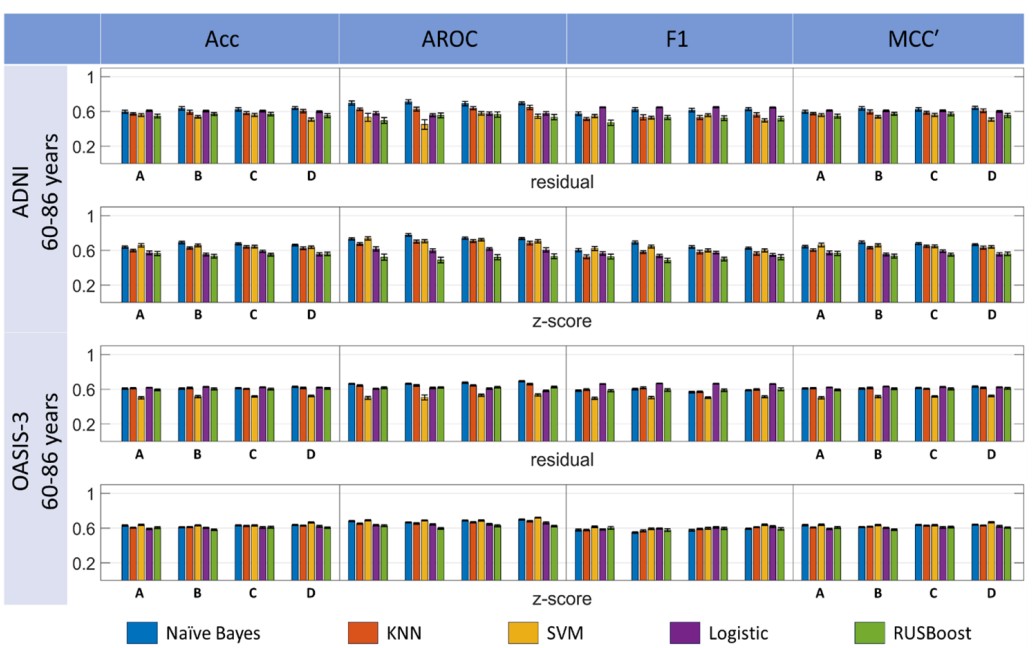

**Figure 5** **Comparison among multiple classification pipeline options, involving five classifiers, four feature selection and two harmonization techniques.** Performance is measured for *randomly balanced cohorts* extracted from ADNI and OASIS-3 imbalanced datasets within a Monte Carlo replication analysis. Results are presented for naïve Bayes, KNN, SVM, Logistic and RusBoost, residual and *z*-score harmonization procedures, as represented in the *x*-axis and legend labels. Bar groups denoted by letters A-D indicate the outcomes corresponding to the different feature subsets that were selected after the MCL app analysis: (A) Features selected using the average scores; (B) Features selected based on the ReliefF criterion; (C) Features selected according to the combination of all evaluated feature selection algorithms; (D) Features selected within the SPSS statistical analysis. Table columns: Acc, accuracy; AROC, area under receiver operating curve; F1, F1-score; MCC', Matthew's correlation coefficient (linearly projected into range [0,1]). Performance metrics are all normalized into the range [0,1] and plotted with a fixed *y*-axis range to enhance visual comparison.

selected in subset D and *z*-score data harmonization, with Acc = 66.58 ± 2.91%, AROC = 72.01 ± 2.40%, and MCC' = 66.78 ± 2.96%. LR performed best according to the F1-score of 66.68 ± 1.21% for ReliefF features and residual harmonization.

When pooling together measures calculated for the four feature subsets for the F1-score, ANOVA with multiple comparison analysis revealed that LR was the best classifier, significantly superior to all the other classifiers for all three datasets using the residual harmonization approach (see Fig. S7 and Table S9). A similar analysis for the MCC' score revealed that naïve Bayes using *z*-score harmonization for the ADNI dataset was superior to the other approaches except for SVM for all three datasets and *z*-score harmonization (see Fig. S8 and Table S10).

## Imbalanced data analysis with customized pipeline

Analogous to the above analysis, Fig. 6 shows the results of the imbalanced data analysis. Here, the divergence among performance metrics is clear. Although the accuracy indicates that SVM may be the best classifier, F1 and MCC' significantly favor RUSBoost at least

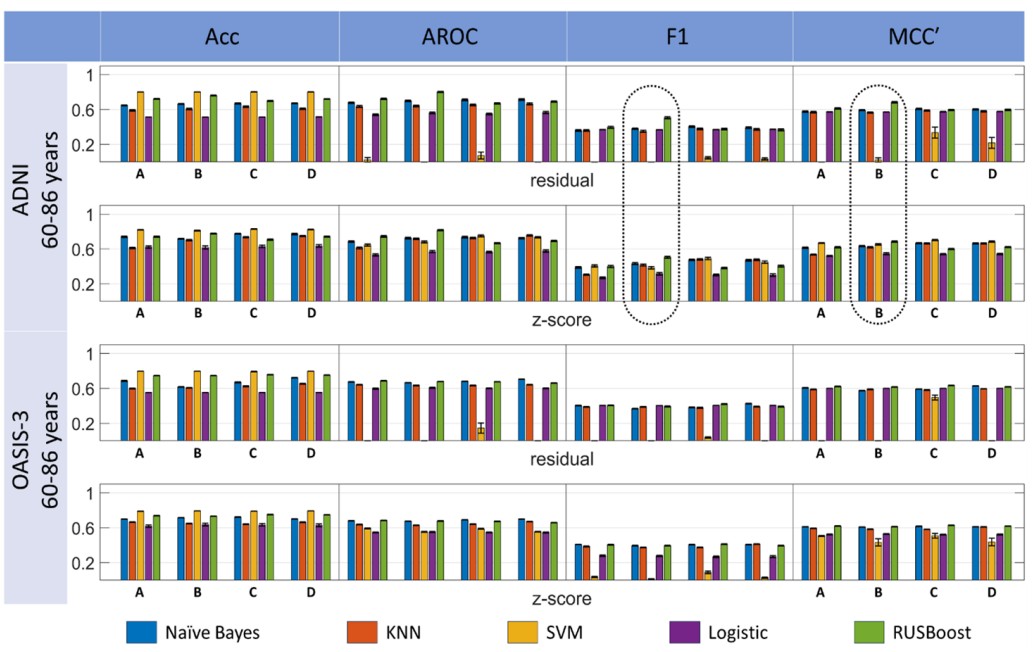

**Figure 6** Comparison among multiple classification pipeline options, involving five classifiers, four feature selection, and two harmonization techniques. Performance is measured directly for ADNI and OASIS-3 imbalanced datasets within a Monte Carlo replication analysis. The ''dotted line'' clips highlight that for ADNI dataset and ReliefF features only, it seems that RUSBoost has a clear advantage over the other classifiers for imbalanced data analysis.

for the ADNI data analysis. With a detailed inspection, we may realize that the accuracy could be biased in this case as the SVM tends to favor the majority group (HC) at the expense of poor classification for the minority group (uHC). Apart from being reflected by the corresponding F1 and MCC scores, this is more clearly visible by inspecting the corresponding true positive rate (TPR) and positive predictive value (PPV) scores, which highlights an overall instability of the SVM classifier in imbalanced data analysis (see Fig. S2 and Tables S3–S4).

For the ADNI imbalanced cohort, RUSBoost achieved the best performance according to two metrics: F1 $= 50.60 \pm 5.20\%$ (based on ReliefF features and residual harmonization) and AROC $= 81.54 \pm 2.92\%$ (based on ReliefF features and $z$-score harmonization). SVM performed best according to the other metrics for subset C features and $z$-score harmonization in both cases: Acc $= 82.93 \pm 1.59\%$ and MCC' $= 70.21 \pm 3.16\%$. For the OASIS-3 age-matched dataset, naïve Bayes showed the best performance according to F1 ($42.54 \pm 1.71\%$, $p_{FDR} < 0.05$) and AROC ($70.33 \pm 1.00\%$; $p_{FDR} < 0.05$), for subset D features and residual harmonization in both cases, with RUSBoost dominating for the MCC' score ($63.31 \pm 1.43\%$), for subset C features and residual harmonization. Here, the accuracy performance was dominated by SVM ($79.58 \pm 0.00$), but this result is invalid, as suspected from the zero-valued error bar. We corroborated that it matches this dataset's percentage of the majority class, *i.e.,* $413 \div (413 + 106) \times 100\%$ (Table 1).

Interestingly, although we employed the default cost error matrix (*i.e.,* $[01;10]$ in order following MATLAB matrix notation) in the balanced data analysis above, we compensated the other classifiers (except RUSBoost) with a customized cost matrix for the imbalanced analysis, penalizing the error committed for classifying a sample in the majority class when the actual class is the critical one: $[01;\delta 0]$ where $\delta$ is the ratio between the cardinalities of the majority and minority groups. When this correction is ignored, the other methods show very poor results. As confirmed by our evaluations, this correction is unnecessary for RUSBoost as it is implemented based on random undersampling (RUS).

For the imbalanced data analysis, when pooling together measures calculated for the four feature subsets for the F1-score, ANOVA with multiple comparison analysis revealed that naïve Bayes using $z$-score harmonization for the ADNI dataset was superior to the other approaches except for SVM for the same combination (see Fig. S9 and Table S11). In contrast, for the MCC' score, the roles were reversed with SVM followed by Naïve Bayes as superior to the rest, also for $z$-score harmonization of ADNI data (see Fig. S10 and Table S12). Remarkably, results for imbalanced were significantly worse than the corresponding ones for balanced analysis, and results were also inferior for the OASIS-3 compared to the ADNI dataset.

## DISCUSSION

In this paper, our primary objective was to develop an MRI-based methodology for early AD prediction, motivated by the fact that MRI is a well-established and widely used technique, providing detailed images for assessing brain regional integrity. This approach enables tracking anatomical changes in the brain during healthy aging and disease progression. In summary, we target the detection of brain changes associated with early cognitive decline by comparing MRI-based features between elders who remained healthy (HC group) and other initially healthy elders who were later diagnosed with mild cognitive impairment (MCI) within 5 years (uHC group), with data provided by ADNI and OASIS-3 longitudinal studies. We presented a ML approach to evaluate multiple feature selection and classification methods. Notably, combining feature selection and statistical analysis approaches, we found that six out of eight significantly detected brain regions in our analyses are consistently reported in the literature as related to early AD anatomical changes: entorhinal, hippocampus, lateral ventricle, lateral orbitofrontal, accumbens area, and middle temporal (see Tables 3–6).

These regions are central to the limbic system's functioning and pivotal in regulating emotions, memory, executive functions, and behavior (*Patestas & Gartner, 2006*). Therefore, anatomical and functional alterations observed in these regions could be critically associated with the early progression of neurodegenerative disorders (*RajMohan & Mohandas, 2007*; *Mori & Aggarwal, 2014*). Notably, the hippocampus and entorhinal cortex (*De Toledo-Morrell et al., 2004*; *Pennanen et al., 2004*) are frequently observed to be affected to a significant degree during the early MCI stage (*Tapiola et al., 2008*). Additionally, changes in the lateral ventricle's size or shape can indicate certain neuro-logical conditions, including neurodegenerative disease, as previously observed in AD

(*Nestor et al., 2008*). Moreover, the orbitofrontal cortex is critical in decision-making, impulse control, and evaluating reward and punishment stimuli. Damage or dysfunction in this area can lead to impairment in these functions and changes in behavior, which are the most common observed symptoms in AD patients, but even in individuals who may receive an MCI diagnosis, as demonstrated by a post-mortem analysis (*Van Hoesen, Parvizi & Chu, 2000*). Moreover, consistent with our results, previous studies have found that the entorhinal cortex is one of the earliest brain regions affected by AD, leading to gradual memory deficits (*De Toledo-Morrell et al., 2004*; *Pennanen et al., 2004*; *Tapiola et al., 2008*). The entorhinal cortex is closely connected to the hippocampus *via* the subiculum, and it is a critical brain region involved in memory formation, spatial navigation, and the processing of associations between different pieces of information.

Our proposed methodology also evaluated the performance of different ML approaches for balanced and imbalanced data analyses using ADNI and OASIS-3 datasets. First, mainly using the MCL app for an accelerated exploration and discovery of "best" method candidates for further analysis, among a vast number of available techniques. This preliminary analysis evaluated the consistency of selected features and classifiers' performance in different conditions. In this analysis, we found that ReliefF (*Robnik-Šikonja & Kononenko, 2003*) consistently outperformed other feature selection techniques, although this superiority was not observed in different scenarios (Tables S1–S4). Interestingly, the stable selection of the same group of brain regions by the different techniques highlighted the importance of the regions mentioned above and our methodology to uncover early AD-linked brain changes.

Although our primary goal is to discover MRI-based biomarkers associated with AD, this should be implemented by exploring and analyzing a wide range of available techniques, as some may be more appropriate than others to reveal important features. In this sense, the MCL app helped to reduce our preselection efforts considerably. This app includes many popular algorithms, such as decision trees, discriminant analysis, logistic regression, naïve Bayes, support vector machines, nearest neighbors, ensemble methods, and neural networks, with predefined and "optimizable" templates. The latter enables selection between different model hyperparameter options and automatic hyperparameter tuning through Bayesian optimization. For example, in the case of neural networks, these options are for the number of layers, number of neurons per layer, and activation function type, among others. Apart from the intensive computational reason, we decided to evaluate "only" all the predefined MCL models since we also assessed in our study a more advanced ML pipeline implementing nested CV with Bayesian optimization for model selection and evaluation, within a Monte Carlo replication framework.

Note also that MCL's above features and classifiers selection analyses may be limited. We exclusively used a unique, randomly balanced cohort extracted from the ADNI dataset since the MCL app's algorithms are optimized for balanced data analysis. However, this is compensated in our study as we selected a wide range of options and used our customized ML pipeline after this preliminary analysis to evaluate selected options further using the original imbalanced ADNI and OASIS-3 datasets. This enabled a more robust evaluation of the combination of five popular techniques (including naïve Bayes, SVM, and

RUSBoost), four feature selection criteria, and two harmonization techniques based on the implementation of nested CV with Bayesian optimization in our pipeline, evaluated within a Monte Carlo replication analysis designed to produce more stable results. The same pipeline was used for balanced and imbalanced data analyses for the same ADNI and OASIS-3 datasets. The unique difference is the implementation of random rebalancing (generating random subsets from the original HC and uHC groups with the same number of samples) within the Monte Carlo analysis before evaluating classification models with the balanced cohorts, as performed for the balanced data analysis. We included RUSBoost in these analyses since it has been reported as one of the best algorithms for imbalanced data analysis (*Seiffert et al., 2010*; *VanHulse, Khoshgoftaar & Napolitano, 2007*). We also implemented a rich set of evaluation metrics, including the F1-score and Matthew's correlation coefficient (MCC), in addition to the traditional accuracy (Acc) and area under receiver operating characteristic curve (AROC) measures because they are deemed more appropriate for imbalanced classification analysis (*Boughorbel, Jarray & El-Anbari, 2017*; *Chicco & Jurman, 2020*). Overall, our study suggests that popular algorithms such as naïve Bayes and LR could be very competitive even for imbalanced data analysis when the algorithm's cost matrix is set conveniently, as in our case, or when random undersampling is considered, as shown with our pipeline implementation (see Figs. 5–6 and discussion therein).

As a warning for future research in this area, all the performance metrics in our study (*e.g.*, Acc, F1, and MCC) showed overfitting (see Figs. S3–S6, in contrast with see Figs. S7–S10), and incorrectly addressing this issue could have negatively impacted our observations. In our case, the use of nested CV helped us to better understand the overfitting effects and achieve more robust outcomes, as noted in the literature (*Varoquaux et al., 2017*; *Varoquaux, 2018*). On the negative side, RUSBoost outcome seems more affected by overfitting as it shows superior performance for imbalanced data analysis based on direct CV measurements (Figs. S5 and S6, for F1 and MCC' scores, respectively). However, this advantage completely disappeared when using the nested-CV (holdout) measurements (see Figs. S9 and S10, respectively, for corresponding score comparisons). This may question the validity of previous results based on RUSBoost for imbalanced data analysis if these studies did not implement a more cautious strategy, like in our case, based on nested cross-validation.

Not least relevant, when comparing directly the balanced analysis *vs.* their imbalanced counterparts for nested-CV measurements (see Figs. S7 *vs.* S9 for visual comparison based on the F1-score and Figs. S8 *vs.* S10 for visual comparison based on the MCC score), it is thought-provoking that balanced analysis produced significantly superior performance results than imbalanced (note the difference in *x*-axis tick range). This observation takes into consideration that both analyses used the same nested-CV pipeline for the same datasets, with the only difference being that balanced analysis applied the nested-CV pipeline for randomly balanced cohorts extracted from the same imbalanced datasets within a Monte Carlo replication analysis (see 'Material and Methods'' for more information; see also Tables 13–14 for more details). This suggests that the rebalancing

approach, evaluated here with our customized pipeline, could improve imbalanced data analysis.

## LIMITATIONS

It is essential to acknowledge the limitations of our study. Primarily, for the balanced data analysis using the ADNI dataset, a smaller sample size can be used to argue for the possible unreliability of the presented results. This small number was mainly due to our consideration of data acquired only with the 3T MRI technique. However, it can also be attributed to the challenge of studying very early AD-linked brain changes. Notice that ADNI also provides data for 1.5T acquired for the very early participants in this study, which we did not consider avoiding this additional confounding factor. However, we can increase the sample size in future studies by including 1.5T and robustly controlling for the possible heterogeneity between 1.5T and 3T MRI images. Additionally, as ADNI is a still ongoing longitudinal study, we can access more data in the future or possibly use different search criteria and methodology to increase the sample size. Another explicit limitation is that the balanced ADNI cohort used in the MCL app analysis may have been selected arbitrarily. We performed a random selection to exclude subjective bias, though a unique ADNI-balanced cohort was mainly used for the feature and classifier selection process. However, this issue appears in many studies that are limited by a small sample size. We compensated for this by selecting a wide range of top-performing options and performing more robust analyses based on the whole ADNI and OASIS-3 imbalanced datasets using our customized ML pipeline. This pipeline enabled balanced and imbalanced data analysis using the same statistical framework combining nested CV and Bayesian optimization. Another explicit limitation is restricting our research to MRI-based AD biomarkers, which we currently address in an ongoing study that includes magneto-electroencephalographic (MEG/EEG) features. Using our pipeline and the findings reported in our study could be valid for more complex analyses involving multimodal neuroimaging features. Lastly, the current research is still far from the goal of developing a quasi-automatic procedure to predict early Alzheimer's disease cases.

## CONCLUSIONS

This study comprehensively compared multiple strategies to identify the most effective AD anatomical biomarkers/features and optimize classifier models for early cognitive decline prediction from MRI data. We identified the predictors through a comprehensive statistical analysis conducted on uncorrected and harmonized data using three different analytical approaches: one-way ANOVA, ANCOVA, and Kruskal–Wallis. Moreover, using the MCL app, we analyzed four feature ranking methods (Chi-square, ANOVA, Kruskal–Wallis, and ReliefF) and multiple classification methods to reduce the number of selected features and classification models for posterior analyses. Subsequently, we used a customized pipeline implementing nested CV and Bayesian optimization to further evaluate the chosen features and classification models within a Monte Carlo replication framework. We enhanced our assessment of the "best" features and models by analyzing

this pipeline's outcome using N-way ANOVA and multiple comparison, along with the Benjamini and Hochberg method to control the false discovery rate methods for assessed performance metrics (*e.g.*, accuracy, F1, and MCC). To ensure the robustness and reproducibility of our results, we validated our methodology using both ADNI and OASIS-3 datasets. Overall, we corroborated that using harmonization approaches improves the evaluation and selection of biomarkers and classification algorithms and that imbalanced data analysis could be improved with ideas such as random rebalancing and nested cross-validation, as implemented with our customized pipeline. Extending our pipeline for use with other multimodal neuroimaging and improving its automatization could be critical for the early detection of Alzheimer's disease and related brain disorders.

## ACKNOWLEDGEMENTS

Data used in preparation of this article were obtained from the Alzheimer's Disease Neuroimaging Initiative (ADNI) database (https://adni.loni.usc.edu/). As such, the investigators within the ADNI contributed to the design and implementation of ADNI and/or provided data but did not participate in analysis or writing of this report. A complete listing of ADNI investigators can be found at: http://adni.loni.usc.edu/wp-content/uploads/how_to_apply/ADNI_Acknowledgement_List.pdf. ADNI is funded by the National Institute on Aging, the National Institute of Biomedical Imaging and Bioengineering, and through contributions from the following: AbbVie, Alzheimer's Association; Alzheimer's Drug Discovery Foundation; Araclon Biotech; BioClinica, Inc.; Biogen; Bristol-Myers Squibb Company; CereSpir, Inc.; Cogstate; Eisai Inc.; Elan Pharmaceuticals, Inc.; Eli Lilly and Company; EuroImmun; F. Hoffmann-La Roche Ltd and its affiliated company Genentech, Inc.; Fujirebio; GE Healthcare; IXICO Ltd.; Janssen Alzheimer Immunotherapy Research & Development, LLC.; Johnson & Johnson Pharmaceutical Research & Development LLC.; Lumosity; Lundbeck; Merck & Co., Inc.; Meso Scale Diagnostics, LLC.; NeuroRx Research; Neurotrack Technologies; Novartis Pharmaceuticals Corporation; Pfizer Inc.; Piramal Imaging; Servier; Takeda Pharmaceutical Company; and Transition Therapeutics. The Canadian Institutes of Health Research is providing funds to support ADNI clinical sites in Canada. Private sector contributions are facilitated by the Foundation for the National Institutes of Health (www.fnih.org). The grantee organization is the Northern California Institute for Research and Education, and the study is coordinated by the Alzheimer's Therapeutic Research Institute at the University of Southern California. ADNI data are disseminated by the Laboratory for Neuro Imaging at the University of Southern California.

### Funding

The data collection and sharing for this project was funded by the Alzheimer's Disease Neuroimaging Initiative (ADNI) (National Institutes of Health Grant U01 AG024904) and DOD ADNI (Department of Defense award number W81XWH-12-2-0012). The

authors were granted access to the Tier 2 High-Performance Computing resources provided by the Northern Ireland High Performance Computing (NI-HPC) facility funded by the UK Engineering and Physical Sciences Research Council (EPSRC), Grant No. EP/T022175/1. Universiti Sains Malaysia and Ministry of Higher Education (MOHE) Malaysia (Scholarship Hadiah Latihan Persekutuan (HLP)) sponsored ALA and Universiti Teknologi PETRONAS supported this study. Roberto C. Sotero was supported by Grant 222300868 from the Alberta Innovates LevMax program. The funders had no role in study design, data collection and analysis, decision to publish, or preparation of the manuscript.

### Grant Disclosures

The following grant information was disclosed by the authors:
Alzheimer's Disease Neuroimaging Initiative: U01 AG024904.
UK Engineering and Physical Sciences Research Council (EPSRC): EP/T022175/1.
Universiti Sains Malaysia and Ministry of Higher Education (MOHE) Malaysia.
Universiti Teknologi PETRONAS.
Alberta Innovates LevMax Program: 222300868.

### Competing Interests

The authors declare there are no competing interests.

### Author Contributions

- Alwani Liyana Ahmad conceived and designed the experiments, performed the experiments, analyzed the data, prepared figures and/or tables, authored or reviewed drafts of the article, and approved the final draft.
- Jose M. Sanchez-Bornot performed the experiments, analyzed the data, authored or reviewed drafts of the article, and approved the final draft.
- Roberto C. Sotero performed the experiments, authored or reviewed drafts of the article, and approved the final draft.
- Damien Coyle conceived and designed the experiments, performed the experiments, authored or reviewed drafts of the article, and approved the final draft.
- Zamzuri Idris conceived and designed the experiments, performed the experiments, authored or reviewed drafts of the article, and approved the final draft.
- Ibrahima Faye conceived and designed the experiments, performed the experiments, authored or reviewed drafts of the article, and approved the final draft.

### Human Ethics

The following information was supplied relating to ethical approvals (i.e., approving body and any reference numbers):

ADNI steering committee and list of acknowledgement for publications using ADNI repository: http://adni.loni.usc.edu/wp-content/uploads/how_to_apply/ADNI_Acknowledgement_List.pdf ADNI protocol and ethics statement: http://adni.loni.usc.edu/wp-content/themes/freshnews-dev-v2/documents/clinical/ADNI-2_Protocol.pdf

### Data Availability

The MATLAB code are available in the Supplementary File.

## Supplemental Information

Supplemental information for this article can be found online at http://dx.doi.org/10.7717/peerj.18490#supplemental-information.

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
