# Peer review of "A machine learning approach for identifying anatomical biomarkers of early mild cognitive impairment"

_PeerJ, doi:10.7717/peerj.18490_

## Round 0.1 · original submission · Major Revisions

With the reviewers' comments in hand, I recommend a major revision. Indeed, even though I think that the manuscript has important strengths, both reviewers highlighted some issues that need to be addressed before its resubmission.

The most pressing concern regards the experimental design. Reviewer 1 finds the methodology unclear and not reproducible, and Reviewer 2 asks for clarification on feature selection and cross-validation details.

Language does not seem to be a major issue, although both Reviewers point towards some room for improvement, and they both lament a lack of clarity and of structure here and there.

Overall, both reviewers, even though with different tones, ask for more details a more clarity concerning the methods and analytic approaches used in the study.

Besides this short summary, I really recommend you engage with each comment raised by the two Reviewers.

Reviewer 1 ·

Basic reporting

The manuscript would benefit from a revision. Indeed, the English needs to be improved, and, in general, the manuscript would need a stronger structure. Sentences are typically too long, some sentences do not have a verb, and wrong sentence connectors are often used (see, for instance, “on the other hand” in line 350). The manuscript is full of information that, however, is not well structured, resulting in a more difficult reading. For instance, in the introduction, a comprehensive description of the literature is reported. Although, this is appreciable, the section is not structured enough. The result is that it is not clear which studies are similar to the proposed one. In this context, a good approach could be starting with describing the literature on the effects of AD on the brain, to focus on the studies using ML and MRI, to finally reporting the ones that discriminate between HC and MCI converted.

Experimental design

The experimental design is not clear and not reproducible, which is a milestone in science. For instance, the authors said they used a "neural network", which is a set of potentially infinite models. The architecture type and the model hyper-parameters (both investigated and selected) should always be reported.

Major comments:
- Participants data section
1) it is not clear whether authors used 1.5T, 3T or both MRI data. This should be clearly stated. From lines 206-207, it seems that only 3T are used, whereas line 208 suggests the use of both data type.
2) what is the purpose of restricting OASIS-3 to a specific age range? How did you choose the age range? Does this have any medical reason? I suppose it is only to match ADNI dataset.

- MRI Pre-processing
1) Half of the paragraph explains what recon-all could do but it is not clear how you used it. Did you segment GM, WM, CSF and all cortical and volumetric areas? Or did you focus on specific brain areas?
2) How many features did you extract?

- Feature and classification model selection
1) Why did you use C_AB for training. The reason you provided is not clear to me, and it seems like an uncommon choice as it is the smallest dataset among the ones you chose.
2) Line 297: “We exclusively used C_AB”. Used for what? Do you mean “for training”?
3) In line 300, authors jump from dataset selection to the predictor variables, which were not mentioned earlier in the manuscript. Are the predictor variables selected somehow? In which way?
4) the description of MRMR is “uses the MRMR algorithm to..”. This does not add any information.
5) Instead of splitting the dataset 10 times, I suggest authors perform stratified-k-fold-validation and repeat it multiple times. This allows us to: 1) keep the class proportions; 2) have an evaluation on a larger testing set; 3) have different splits to assess model stability to data splitting. Following your approach, you will inevitably end up with a large accuracy variance, which is not informative. By using this approach, the variance among different stratified-k-fold-validation results provides a measure of model stability. Importantly, splitting should always be done with fixed random states to ensure reproducibility.
6) If I understood well, you first chose the “best” model and then looked for the best hyper-parameters of that model. If this is the case, normally you do exactly the opposite: you look for the best hyper-parameters, and then you pick the best model. Otherwise it is not a fair comparison.
7) In line 352, the authors claim that the described procedure is applied to all six datasets. So it seems to be that 6 different models have been trained, which is not the same impression I had previously in the manuscript.

Validity of the findings

results could potentially be interesting, as multiple analysis have been carried out. However, as the methodological part is unclear, it is hard to evaluate the validity of the findings.
As the authors also claimed, accuracy is not a good metric for model evaluation in the presence of class imbalance. I suggest the use of the f1-score, which is the most commonly adopted metric in these contexts.

Additional comments

HC and uHC abbreviations are introduced continuously in the manuscript. Please, do it just once. The same criteria holds for all the other abbreviations, such as ICV. On the contrary, “machine learning” is repeated multiple times in the text. I suggest using the abbreviation ML.

Line 89: “first AD symptoms” or “first symptoms of AD”. The subject (it) is missing.
Line 189: “and” seems to be linked
Line 201: “that is, mainly […]” does not contain a verb. Link this sentence to the previous one or make it a proper sentence.
Line 203: what is the “ADNIMERGE excel table”? If you mention it, you should either report on your manuscript or cite it properly (e.g., provide the link to it).
Line 223: the expression “original age range for balanced data” is not clear
Line 225: “restricted age range” to which age range?
Line 265: Although officially used in the ADNI dataset, do not use the notation “normal control” if you refer to them as healthy controls in the manuscript.
Line 265: authors states “we used the whole dataset from NC, MCI and AD groups”. Why do you use AD group if your purpose is to discriminate between healthy and MCI-converted subjects?
Line 267: from this sentence, it seems you only used features from the hippocampus. If this is the case, this information should be reported much earlier in the text.
Line 287: there is a typo reporting a red strikethrough “s”
Line 289: it seems you fine-tuned the models but this was not mentioned earlier as adopted approach.
Line 299: what do you mean by “ameliorate double dipping”
Line 320: “rankS the features”. Same holds for line 308 if “rank” is used as verb.

Abstract
- the sentence in Objective is too long and not reader friendly
- authors claim that they performed the same analyses SEVERAL times for OASIS-3 but, right after, they only report 2 experiments (i.e. firstly […], and secondly […]).
- in “Methods”: HC and uHC were already introduced previously. Please, introduce the notation once and be consistent with it.
- “Then, the selected imbalanced and balanced OASIS-3 cohorts by 53 restricted this dataset to the same age range as ADNI (60 ñ 86 years old).” this sentence seems to miss the verb
- The last sentence in “Conclusion”, referring to the potential future integration with MEG/EEG data, is completely unrelated from the previous part. Data integration is not mentioned above, your method is not easily transferable to data integration, and the choice to mention MEG/EEG data rather than PET, genetic, clinical data is not clear.
- Do not write “Alzheimer’s” but Alzheimer’s Disease or its abbreviation AD

Generally speaking, the abstract should go under important revision as it is not easy to read. Mostly important, in methods section, authors should add the essential information, including: I) which data has been used (1.5 – 3T); ii) how pre-processing has been carried out, which region of interest (if any) has been selected and which features have been extracted; iii) how many and which machine learning classifiers have been used. Further, by reading the method part, it is not clear on which dataset training has been performed (i.e., if you replicated your method on both dataset or tested the method generalization on multiple datasets).

Reviewer 2 ·

Basic reporting

The manuscript is written in good English, with occasional typos or language issues. For example, dangling sentences at lines 52-53 ("Then, the selected...") and 578 ("Thus, allowing the tracking..."); check "tress" (line 140), "ReliefT" (line 141), "performances" (line 293).

Introduction provides sufficient background information and references.

Make sure the abstract subheadings are followed by a period, as per the journal's Standards.

In general, the Figures are HQ and appropriately labelled and described.
- Figure 2, A-C: consider using transparency for the points and increasing line widths to improve readability. Clarify "data1-6" in panel C legend.
- Figure 5: usually, more than 6 series in a plot make it difficult to read. Consider selecting a subset of model x feature selection x normalization for this figure and migrating the rest to Supplementary information.

Experimental design

I find the methodology and experimental design appropriate overall, with a few points that require further clarification.

It is not clear whether feature selection is performed solely on ADNI cohort C_AB (line 297). Please clarify if you use all the other cohorts but C_AB in the subsequent steps of the analysis (e.g., line 332, "next, in the MCL app, we split the data into a train (80%) and test (20%) subsets ..."). If not, feature selection should be performed only on the training split to avoid information leakage from the test set.

Please specify k for k-fold CVs used in the analyses (e.g. lines 287-288, 342), and whether the class label proportions were preserved in each partitioning operation (CV folds, holdout, 80/20% split).

Bonferroni correction is quite conservative. Consider an alternative such as the Benjamini and Hochberg method, which controls the false discovery rate.

It is well known that accuracy and ROC AUC may be optimistic in the context of imbalanced domains. I suggest complementing the reported classification metrics with alternatives that are more robust in this scenario, such as F1-score, Matthews Correlation Coefficient, and the area under the precision-recall curve (AUPRC).

Validity of the findings

Discussion is appropriate and also points out the current limitations of the study.

Additional comments

No comments.

---

## Round 0.2 · Minor Revisions

As you can see, both the reviewers are almost satisfied with your resubmitted version, although they raise some minor points that should be addressed.

Reviewer 1 ·

Basic reporting

LANGUAGE:
The English is fine, with few mistakes that should be taken into account.
Line 38: “to classify between” is not gramatically correct. You can say “to classify into” or “to discriminate between”.
Lines 175 , 222, 270, 473: Please use sex instead of gender.

REFERENCES:
All methods should be introduced or cited to inform the reader about their functionality. For instance, ReliefF (line 137) , Svm (line 131), Ripper (line 146), etc. Please add a reference.
Same think holds for line 177 referring to SPSS statistical software.


ARTICLE STRUCTURE:
The general form and the structure of the manuscript have been greatly improved.

Authors’ contribution: by reading the introduction, it should be clear what the contribution of the work is. Line 154 introduces your study but, however, it does not tell anything about methodology. Line 155 states the consequences of your study rather than the content. Authors should add few lines in which they explain what their method consists in. Similarly, in the Abstract (line 47), the sentence “monte carlo replication analysis was used for imbalance and randomly balanced datasets” is not clear. I suggest explicitly stating what this method is used for.

Experimental design

ADNI DATASET
Line 41: ADNI stands for “AD neuroimaging initiative”
Please, add the ADNI phase from which data is downloaded.

EXPERIMENTAL SETUP:
Line 225: authors claim that ADNI is used as primary data. This would suggest that it has been employed for classification training. However, authors state “mainly for feature selection and evaluation of ML classifier”. This is confusing. Please use the expression “evaluation of classifiers” ONLY to refer to the testing phase.
Line from 379 to 387: From this paragraph, I understand that adni-balanced is used for training and evaluation, whereas adni-imbalanced and OASIS-3 are for evaluation only.
From the results section, it seems instead that after feature and classifier selection, the classifiers are re-trained on imbalanced ADNI and OASIS-3. Again, I think that the main issue is that authors do not use technical expressions from the ML field but they use generic verbs, such as "to apply" and "to evaluate" which have, however, a different meaning in ML. When saying "to evaluate a ML classifier", the ML community means that the classifier has been previously trained on another dataset and the specific dataset of interest has been used for testing only. Authors should explicitly state on which set the classifiers have been trained. I think that the article would benefit from the contribution of an expert in ML.

Use of neural network
Authors report NN information in the Discussion section. This is not the appropriate section. Model details must be included in Material and

Validity of the findings

The experimental pipeline is solid.

Additional comments

Lines 132 and 430: the line starts with the dot.

Reviewer 2 ·

Basic reporting

The reviewed manuscript has improved substantially (abstract, introduction and background, methodology, figures, English). There seems to be an issue with the figure captions, which are truncated (probably this is related to the submission system).

Experimental design

I thank the Authors for considering my suggestions. I support the Authors' choice of using the normalized MCC so that all metrics are in the (0, 1) range. In the new section "Classification performance metrics", I advise you to refer to it as "normalized MCC" rather than "modified MCC". Moreover, in the Abstract (line 50), please introduce the normalized MCC directly, since later in the abstract you report results in terms of MCC'.

Validity of the findings

My previous comments were addressed. No further comments.

Additional comments

No further comments.

---

## Round 0.3 · accepted · Accept

Dear Authors,

I can see that you properly addressed the minor points raised in the last round of review. Therefore, I am glad to inform you that I now consider your manuscript suitable for publication on PeerJ.